# Dysregulated cholesterol homeostasis results in resistance to ferroptosis increasing tumorigenicity and metastasis in cancer

Wen Liu[1], Binita Chakraborty[1], Rachid Safi[1], Dmitri Kazmin[2], Ching-yi Chang[1] & Donald P. McDonnell [1✉]

Hypercholesterolemia and dyslipidemia are associated with an increased risk for many cancer types and with poor outcomes in patients with established disease. Whereas the mechanisms by which this occurs are multifactorial we determine that chronic exposure of cells to 27-hydroxycholesterol (27HC), an abundant circulating cholesterol metabolite, selects for cells that exhibit increased cellular uptake and/or lipid biosynthesis. These cells exhibit substantially increased tumorigenic and metastatic capacity. Notably, the metabolic stress imposed upon cells by the accumulated lipids requires sustained expression of GPX4, a negative regulator of ferroptotic cell death. We show that resistance to ferroptosis is a feature of metastatic cells and further demonstrate that GPX4 knockdown attenuates the enhanced tumorigenic and metastatic activity of 27HC resistant cells. These findings highlight the general importance of ferroptosis in tumor growth and metastasis and suggest that dyslipidemia/hypercholesterolemia impacts cancer pathogenesis by selecting for cells that are resistant to ferroptotic cell death.

---

[1] Department of Pharmacology and Cancer Biology, Duke University School of Medicine, Durham, NC 27710, USA. [2] Emory Vaccine Center, Emory University, Atlanta, GA 30322, USA. ✉email: donald.mcdonnell@duke.edu

Obesity, dysregulated lipid homeostasis, and other sequelae of the metabolic syndrome are associated with increased risk of breast cancer and with poorer outcomes in patients with established disease[1,2]. Of late, there has been substantial interest in defining the specific impact of cholesterol on breast cancer pathogenesis as hypercholesterolemia is a risk factor that can be modified by diet and/or pharmaceutical intervention[3]. However, the results of a large number of epidemiological studies have highlighted the complexity of cholesterol biology in cancer. These studies report different effects of total, LDL, and HDL-cholesterol on cancer incidence, the magnitude of which can be influenced by variables such as estrogen receptor (ER) and menopausal status, disease subtype, and comorbid inflammatory conditions[3–6]. There is considerable evidence that hypercholesterolemia is an independent risk factor for breast cancer in postmenopausal women[3]. However, studies that have evaluated the impact of statins (the most commonly used LDL-cholesterol-lowering drug) on breast cancer risk in this population have yielded equivocal results[7]. More compelling are the results of several large studies that demonstrate significant benefits in overall and disease-free survival in patients taking a statin before or following a cancer diagnosis[8–13]. Understanding the different effects of hypercholesterolemia on disease incidence versus established disease will inform the development of new approaches to modify this axis for maximal therapeutic benefit.

There is abundant experimental evidence linking dysregulated cholesterol homeostasis to cancer pathobiology. This is not surprising as cholesterol serves as a fundamental building block for cell membranes, and the ability of cells to increase the synthesis or accumulation of cholesterol is a prerequisite for proliferation[14]. It has also been demonstrated that alterations in the cholesterol content of membranes can modulate lipid raft formation and indirectly influence the activity of the smoothened receptor (hedgehog signaling)[15,16], adenosine A2A receptor[17], LDLR-related protein 5/6 receptors (Wnt signaling)[18], and also mTORC1 signaling[19]. Increased intratumoral cholesterol may facilitate local steroidogenesis, although direct demonstration of this in vivo is lacking[20,21]. Changes in the abundance of metabolites within the cholesterol biogenesis pathway, including mevalonate, isoprenoids, ubiquinone, and oxysterols, have also been shown to regulate processes important in cancer pathogenesis[22]. Notably, we and others have determined that 27-hydroxycholesterol (27HC), an abundant oxysterol and a primary metabolite of cholesterol, functions as a bona fide endogenous SERM (selective estrogen receptor modulator), supporting the growth of ER-positive tumors in several models of luminal breast cancer[23,24]. It appears likely, therefore, that some of the benefits of statins on the survival of postmenopausal breast cancer patients with ER-positive disease is attributable to their ability to reduce 27HC production. Independent of ER, elevated cholesterol/27HC influences breast cancer metastasis in animal models in a cancer cell-extrinsic manner by stimulating CXCR2-dependent recruitment of polymorphonuclear-neutrophils and γδ T-cells to the metastatic niche[25]. The importance of this pathway in human tumor biology remains to be determined.

Studies from our group and others have shown that the circulating levels of 27HC correlate well with total cholesterol in humans and in animal models and that statin dependent reductions in total/LDL-cholesterol result in a commensurate reduction in 27HC[23–27]. 27HC is primarily produced in the liver by the monooxygenase sterol 27-hydroxylase (CYP27A1) where it can be further metabolized to more polar, readily secreted, metabolites by 7-alpha-hydroxylase (CYP7B1)[28,29]. However, the observation that both enzymes are also expressed within breast tumors suggests that both the systemic and local production of 27HC may impact tumor biology[23,24]. In ER-positive tumors, it

has been shown that, whereas CYP27A1 expression does not vary much between tumors, a significantly reduced expression of CYP7B1 was noted, resulting in increased intratumoral levels of 27HC[24]. Regardless of the source; however, it is clear that 27HC can directly promote tumor growth by functioning as an estrogen and can indirectly impact tumors by attenuating the efficacy of endocrine therapies[23,24,30,31].

In contrast to what is seen in ER-positive tumors, CYP27A1 is overexpressed in ER-negative tumors and this is associated with a higher tumor grade and poorer outcomes[23,26]. This latter finding has been difficult to reconcile with what is known about the ER-independent function(s) of 27HC as a negative feedback regulator of sterol regulatory element-binding protein (SREBP1/2) activity, the master regulator of the cholesterol biosynthesis pathway[32,33]. Specifically, 27HC interacts directly with INSIG1/2 in the endoplasmic reticulum[34], which in turn blocks SCAP-dependent activation of SREBP2 resulting in decreased expression of 3-hydroxy-3-methylglutaryl (HMG)-CoA reductase (HMGCR), and the low-density lipoprotein receptor (LDLR). In addition, 27HC also acts as a ligand for the liver X receptors (LXR) and, as such, promotes cholesterol efflux through the ATP-binding cassette transporters A1 and G1 (ABCA1 and ABCG1)[35,36]. Considering this biology, it was unclear how increased expression of CYP27A1, and increased production of 27HC from cholesterol, could contribute negatively to the pathobiology of ER-negative tumors and addressing this apparent paradox was the primary goal of this study.

In this study, we demonstrate that chronic exposure of cancer cells to 27HC, that which likely models the situation in patients with hypercholesterolemia/dyslipidemia, results in the emergence of cells exhibiting increased tumorigenic and metastatic capacity. In responding to 27HC cells increase their uptake and/or synthesis of lipids. The metabolic stress associated with increased lipid accumulation requires the expression of the lipid peroxidase GPX4, a key negative regulator of ferroptotic cell death. Importantly, we show that resistance to ferroptosis is a feature of metastatic cells and further demonstrate that GPX4 knockdown attenuates the enhanced tumorigenic and metastatic activity of cells chronically exposed to 27HC. These findings highlight the general importance of ferroptosis in tumor growth and metastasis and suggest that dyslipidemia/hypercholesterolemia impacts cancer pathogenesis by selecting for cells that are resistant to ferroptotic cell death.

## Results

### The oxysterol 27-hydroxycholesterol attenuates cancer cell growth through its ability to inhibit cholesterol uptake/synthesis.
Previously, we and others have shown that the oxysterol 27HC, produced from cholesterol by the monooxygenase CYP27A1, can function as a partial agonist of the estrogen receptor (ER). It was further demonstrated that the estrogenic activity of 27HC was sufficient to support the growth of ER-positive breast tumors in several relevant mouse models, suggesting that it serves as a biochemical link between hypercholesterolemia and breast cancer pathobiology. This mitogenic activity of 27HC was unexpected given its normal physiological role as a negative feedback regulator of cholesterol biosynthesis/import through its inhibitory actions on INSIG1/2, an obligate cofactor for SREBPs. This suggested that the actions of 27HC on ER may attenuate its inhibitory activity on cholesterol uptake/synthesis and that in ER-negative cells, 27HC may function differently, a hypothesis we addressed in these studies.

For comparative purposes, we first assessed the activity of 27HC on the growth of ER-positive MCF7 cells in media stripped of exogenous steroids (charcoal-stripped media (CFS)) versus

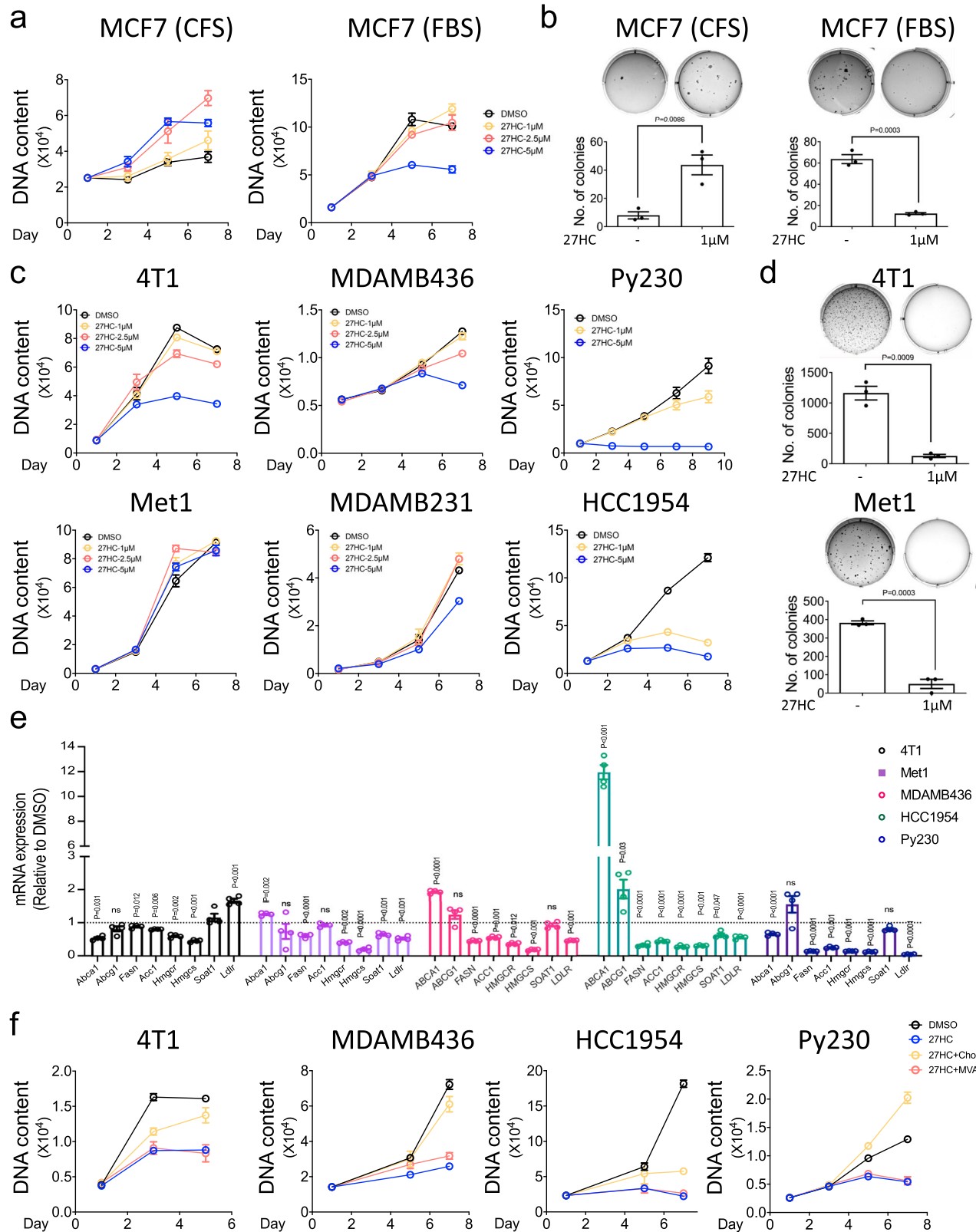

complete media. For these studies, we used a range of 27HC concentrations from those found in the circulation of normo-cholesterolemic individuals (~1 μM) to those found in patients with hypercholesterolemia (2.5–5 μM)[36–40]. Consistent with our previous findings it was observed that in CFS, the estrogenic activity of 27HC was sufficient to support cell proliferation.

However, when the same studies were performed in FBS, 27HC inhibited cell proliferation to a level consistent with its inherent partial ER agonist activity (Fig. 1a). Similarly, 27HC was shown to support colony formation of MCF7 cells in CFS and inhibit colony formation in FBS, an expected manifestation of its ER partial agonist activity (Fig. 1b). These data are in contrast to

**Fig. 1 27HC inhibits the growth of ER-negative breast cancer cells. a** ER-positive MCF7 cells were plated in charcoal stripped serum (CFS, left) and complete (FBS, right) media, followed by treatment with the indicated doses of 27HC. Cells were then harvested at different time points and cell growth was assessed by measuring DNA content using Hoechst 33258. **b** MCF7 cells were seeded in 6-well plates in soft agar mixed with media containing stripped (CFS, left) and complete serum (FBS, right), followed by the treatment with vehicle (0.1% DMSO) or 27HC (1 μM), and incubated for 3 weeks. Colonies were stained with crystal violet. The graph represents the number of colonies growing in soft agar per well. **c** Time-dependent cell growth curves of various ER-negative breast cancer cells (4T1, Met1, MDAMB436, MDAMB231, Py230, and HCC1954) treated with indicated concentrations of 27HC. **d** Soft agar assay showing that 27HC (1 μM) inhibits the colony formation of 4T1 and Met1. **e** qRT-PCR analysis of the mRNA expression levels of LXR, SREBP1c, and SREBP2 (genes involved in lipid metabolism) from 4T1, Met1, MDAMB436, HCC1954, and Py230 cells treated with 27HC (5 μM) for 24–72 h. **f** Supplementation with cholesterol (Chol, 10 μM) but not mevalonate (MVA, 500 μM) reversed the antiproliferative effects of 27HC (5 μM) in 4T1, MDAMB436, HCC1954, and Py230 cells. Data are plotted as mean ± SEM as representative results from two to four independent experiments (except Fig. 1b which was performed a single time); $n = 5$ wells of cells (a, c, and f); $n = 3$ wells of cells **b** and **d**; $n = 4$ wells of cells **e**. *P* values were calculated using two-sided unpaired Student's *t* test **b**, **d**, and **e**. Numerical source data are reported in the Supplementary Data 1 and in the Source Data File.

what we observed in ER-negative cell lines, where 27HC was shown to inhibit the proliferation of the 4T1, MDAMB436, Py230, MDAMB231, and HCC1954 breast cancer cell lines (Fig. 1c) and the BPD6 and B16F10 cellular models of melanoma (Supplementary Fig. 1a) in regular media containing lipids. When cultured in 2D, estrogen unresponsive Met1 cells were shown to be resistant to the antiproliferative activity of 27HC. However, 27HC effectively inhibited the ability of 4T1, Met1 (Fig. 1d) BPD6, and B16F10 cells (Supplementary Fig. 1b) to form colonies in a 3D colony formation assay. 27HC also inhibited the migration of all ER-negative cells examined (Supplementary Fig. 1c). Together these data suggest that, unlike the situation in ER-positive cells, in ER-negative cells 27HC functions to inhibit cell proliferation, colony formation, and in vitro migration.

We considered it likely that the effects of 27HC on the biology of ER-negative cancer cells observed could be attributed to its established role as a negative regulator of cholesterol synthesis/uptake through its actions on both the SREBP and LXR dependent signaling pathways. Indeed, we found that in the cancer cells examined, with some subtle differences noted between cells, 27HC inhibited the expression of SREBP1c and SREBP2 target genes with evidence of LXR agonist activity also being observed in some cells (Fig. 1e, Supplementary Fig. 2a and Supplementary Data 1). We do not believe that the LXR activity of 27HC contributes significantly to its actions on cancer cells as we have shown (a) that it is a relatively weak partial agonist of LXR in the cells we have explored (Supplementary Fig. 2b) and (b) full agonists of LXR (i.e., GW3965) do not phenocopy the antiproliferative activity of 27HC in these cells (Supplementary Fig. 2c). Thus, whereas 27HC can activate LXR and induce the expression of genes involved in cellular cholesterol efflux, this activity is not sufficient to inhibit cancer cell proliferation. This suggests instead that inhibition of SREBP-dependent lipid/cholesterol uptake and synthesis may be the primary function of 27HC. In support of this idea, we demonstrated that cholesterol supplementation reverses both the antiproliferative activity of 27HC (Fig. 1f) and its ability to inhibit colony formation (Supplementary Fig. 2d). The inability of mevalonate and other upstream intermediates of cholesterol biosynthesis to rescue 27HC inhibition of cancer cell growth or colony formation (Fig. 1f and Supplementary Fig. 2e, f) suggests that the primary action of 27HC in these cells is to inhibit cholesterol synthesis/uptake and that this likely explains its observed antiproliferative activity.

**Chronic exposure to 27HC selects for cells that are more tumorigenic and metastatic when evaluated in vivo.** The observation that 27HC inhibits the growth of ER-negative breast cancers likely due to its ability to disrupt cholesterol homeostasis, while somewhat expected, was difficult to understand in light of the established links between elevated cholesterol (and 27HC)

and cancer progression, and the observation that elevated expression of CYP27A1 is associated with the most aggressive, high grade, ER-negative tumors. We considered that while these clinical observations are inconsistent with our observation that 27HC inhibits cell proliferation, it was also possible that it could relate to the way hypercholesterolemia was modeled in our studies by subjecting cells/tumors to acute elevations in cholesterol/27HC. Hypercholesterolemia is a chronic disease and we reasoned that the impact of acute and long-term exposure to elevated 27HC might have different effects on cancer cell biology. To explore this possibility, we created 27HC resistant versions of 4T1, Py230, BPD6, HCC1954, MDAMB436, and B16F10 cells by passaging them in the continued presence of 5 μM 27HC for 1–4 months in 2D culture until colonies emerged. The concentration of 27HC used was that reported previously in hypercholesterolemic patients and mouse models of dyslipidemia[37,40]. The time/dose-response growth curves of the 27HC sensitive and isogenic resistant variants of 4T1, Py230, and BPD6 cells propagated in 2D culture are shown in Fig. 2a and Supplementary Fig. 3a. Whereas dramatic differences in sensitivity to 27HC were noted (27HCS versus the 27HCR cells), the overall proliferative capacity of the sensitive and resistant cells was similar in the absence of 27HC. However, when propagated as xenografts in mice, the 27HC resistant derivatives of all of the three cell models were shown to be substantially more tumorigenic than their 27HC sensitive counterparts (Fig. 2b). Particularly striking was the observation that when propagated in vivo, parental 27HC sensitive Py230 (27HCS) cells gave rise to small slow-growing tumors, whereas their 27HCR counterparts readily formed tumors[41]. Similarly, two different 27HC resistant variants of the HCC1954 human breast cancer cell model were found to be substantially more tumorigenic than their 27HC sensitive derivatives (Supplementary Fig. 3b). In these in vivo studies, the tumors were propagated in mice in the absence of exogenous 27HC, suggesting that the increased tumorgenicity observed was not a result of a direct mitogenic action of 27HC but to a collateral activity that is acquired as cells become resistant to the antiproliferative activities of 27HC.

The in vitro migratory activity of all the 27HCR cells examined was substantially greater than their sensitive counterparts (Supplementary Fig. 3c). Since migratory capacity in vitro is a surrogate for metastasis, we assessed the lung metastatic capacity of the sensitive and resistant cells using an established tail vein injection model. In all three models examined (4T1, Py230 and BPD6) resistance to 27HC was associated with dramatically increased metastatic potential (Fig. 2c). Taken together, it appears that cells, in adapting to increased levels of 27HC, acquire activities that render them more tumorigenic, and metastatic.

**Increased intracellular lipid accumulation is a characteristic of resistance to 27HC and is causally linked to tumorigenicity.**

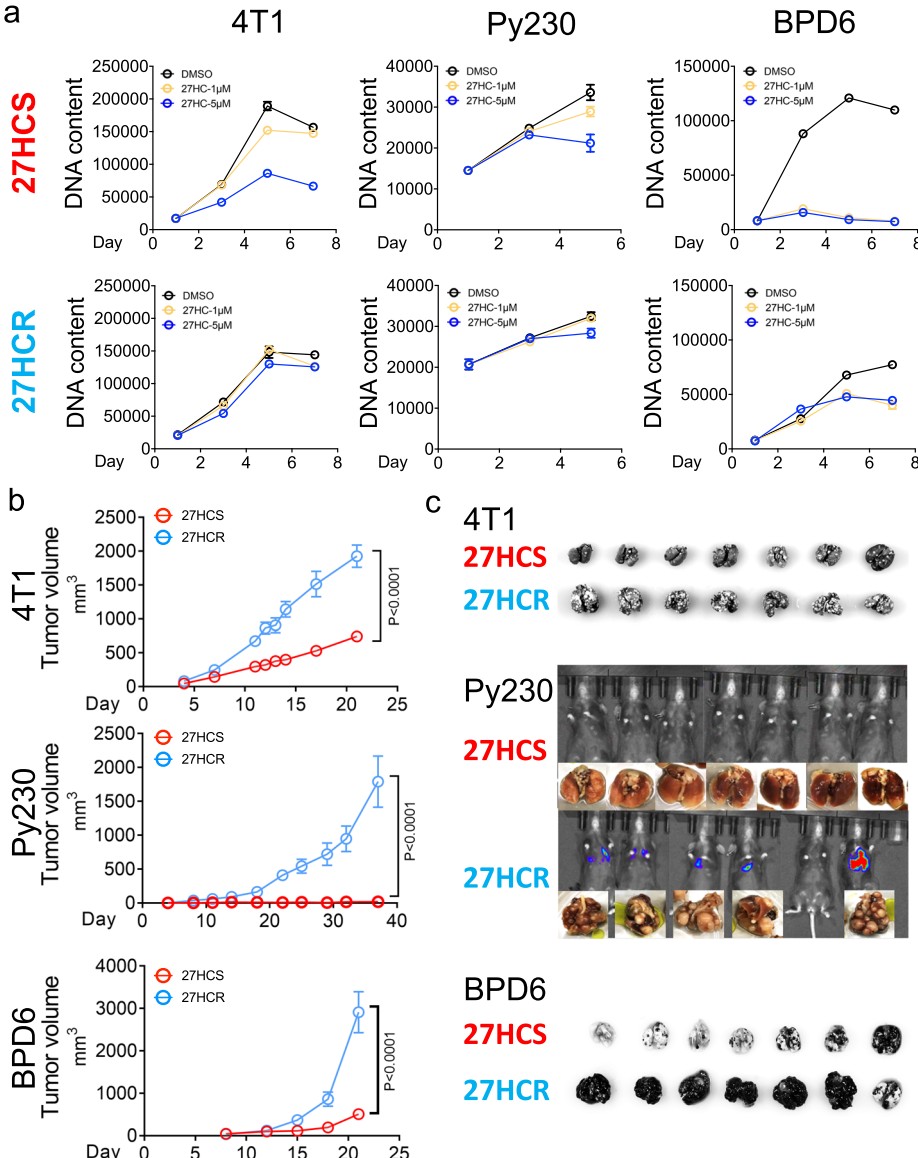

**Fig. 2 Chronic exposure to 27HC selects for cells with increased malignant phenotypes. a** ER-negative cancer cells were treated with 0.1% DMSO or 27HC (5 μM) for 1–4 months. Emergent cells are resistant to the antiproliferative effects of 27HC. Representative data from 4T1 (left), Py230 (middle), and BPD6 (right) cancer models are shown. Data plotted as mean ± SEM as representative results from three independent experiments, n = 5 wells of cells. **b** Xenograft tumors from 27HCS and 27HCR cells in 4T1 (upper), Py230 (middle), and BPD6 (lower) in syngeneic mouse models. (4T1-27HCS, n = 8 mice; 4T1-27HCR, n = 8 mice; Py230-27HCS, n = 6 mice; Py230-27HCR, n = 8 mice; BPD6-27HCS, n = 8 mice; BPD6-27HCR, n = 8 mice), Data plotted are mean ± SEM, P values were calculated using two-way ANOVA. **c** Lung metastasis from 27HCS- and 27HCR cells in 4T1 (upper), Py230 (middle), and BPD6 (lower) intravenously injected in syngeneic mouse models. Mice were euthanized on Day 21 (4T1), Day 78 (Py230), and Day 20 (BPD6), respectively. Numerical source data are reported in the Source Data File.

The biochemical mechanisms underlying resistance to 27HC and how they contribute to the increased tumorigenicity and metastatic capacity of cancer cells were next assessed. Most of the exploratory mechanistic studies were performed in the 4T1, Py230 and HCC1954 breast cancer cell models but key findings were confirmed in other cells. We first probed the possibility that the mechanisms that enable 27HC to downregulate cholesterol/lipid synthesis may have been disrupted in the resistant cells. However, gene expression analysis demonstrated little change in the mRNAs encoding the key regulatory steps in these processes under basal conditions, and it was observed that in both sensitive and resistant cells, 27HC remains an effective inhibitor of their expression (Fig. 3a, Supplementary Fig. 4a and Supplementary Data 2). These data indicate that in the resistant cells the negative

feedback by 27HC on SREBP activity remains intact. It was noticed, however, using BODIPY staining to assess lipid content, that all 27HCR cells examined accumulate significant amounts of neutral lipids stored as cytoplasmic lipid droplets (LDs) (Fig. 3b and Supplementary Fig. 4b). In general, LDs consist of a mono-layer of phospholipids surrounding a neutral lipid core comprised of triacylglycerols (TAGs) and cholesteryl esters (CEs)[42]. This suggested to us the possibility that in order to counteract the inhibitory effects of 27HC on cholesterol and lipid synthesis, 27HCR cells may increase their ability to uptake neutral lipids. To address this directly, and to confirm the identity of the accumulated lipids, we performed a comprehensive lipidomics analysis in 27HCR and 27HCS derivatives of HCC1954 cells. Because exogenous lipids can impact cellular response to lipid

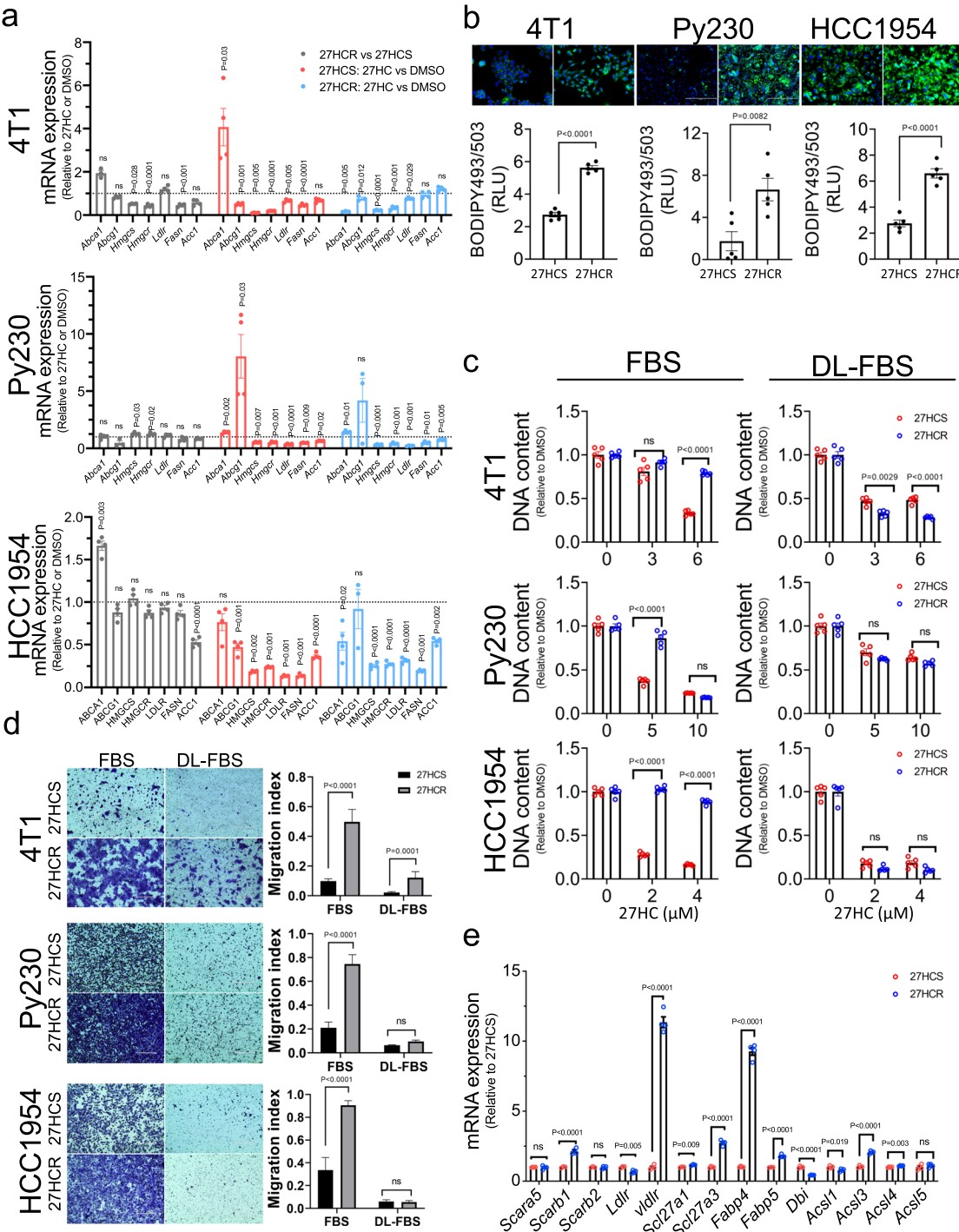

**Fig. 3 Increased lipid uptake is a feature of cells resistant to the antiproliferative actions of 27HC. a** qRT-PCR profiling of the expression of select genes involved in lipid metabolism in 27HCS and 27HCR cells treated with 0.1% DMSO or 27HC (5 μM) for 24–72 h. Representative data from 4T1, Py230, and HCC1954 cancer models are shown. Data are plotted as mean ± SEM as representative results from two (HCC1954), four (Py230), and one (4T1) independent experiments, n = 4 wells of cells. **b** Lipid droplet content in 27HCS and 27HCR derivitives of 4T1, Py230, and HCC1954 cells were visualized using BODIPY 493/503 staining. Data are plotted as mean ± SEM as representative results from three independent experiments, n = 5 wells of cells **c** 27HCS and 27HCR derivatives of 4T1, Py230, and HCC1954 cells were cultured in lipid-rich (FBS) and lipid-depleted (DL-FBS) serum-containing media and treated with 0.1% DMSO or 27HC at indicated doses, followed by assessment of cell growth. Data are plotted as mean ± SEM as representative results from three independent experiments, n = 5 wells of cells **d** Migration assays performed using 27HCSand 27HCR derivatives of 4T1, Py230, and HCC1954 cells cultured in lipid-rich (FBS) and lipid-depleted (DL-FBS) serum-containing media. Data are plotted as mean ± SEM; n = 8-20 random fields measurements from a total of three transwell chambers. **e** qRT-PCR analysis of the mRNA expression levels of genes involved in lipid uptake and trafficking in 27HCS and 27HCR derivatives of 4T1, Py230, and HCC1954 cells. Representative results from Py230 cells are shown, and results for other cell lines are shown in Supplementary Data 4. Data are plotted as mean ± SEM as representative results from two (4T1 and HCC1954) and eight (Py230) independent experiments; n = 4 wells of cells; P values were calculated using a two-sided unpaired Student's t test **a**, **b**, and **e** and one-way ANOVA with Tukey's post hoc test **c** and **d**. Numerical source data are reported in the Supplementary Data 2 and in the Source Data File.

perturbation, we extracted lipids from HCC1954-27HCS and −27HCR cells growing in both FBS (lipid rich) and charcoal-stripped (lipid poor) media. This analysis revealed that HCC1954 cells contain abundant phospholipids (PL, 53.8%) and neutral lipids (NL, 24.27%) with sterols representing a low percentage of the overall lipid content (2.63%) (Supplementary Fig. 4c). Importantly, in lipid-rich media, the total lipid content in two different 27HC resistant cell lines (HCC1954-27HCR1 and -27HCR2) was significantly increased over that in HCC1954-27HCS cells grown under the same conditions (Supplementary Fig. 4d). Indeed, the abundance of all lipids were increased in cells grown in lipid-rich media but not lipid-poor media (with the exception of neutral lipids), indicating that increased lipid uptake is a feature of 27HCR cells in this model. Neutral lipids were increased in the HCC1954-27HCR cells in a manner that was independent of media lipid content, suggesting that these cells may also have the capacity to increase de novo lipid synthesis as they develop resistance. Interestingly, when compared to what was seen in 27HCS cells, cholesterol levels were increased in 27HCR cells when grown in lipid-rich media. In lipid-poor media no differences in cholesterol content were noted between cells although overall cholesterol content was increased. (Supplementary Fig. 4e). Further analysis of the lipid profiles in the sensitive and resistant cells revealed that neutral lipids, of which the majority contain monounsaturated lipids, were the most significantly increased in the 27HCR cells (Supplementary Fig. 4f–g and Supplementary Data 3). Specifically, the levels of total monounsaturated and oleic acid-containing neutral lipids are significantly higher in 27HCR cells when compared to 27HCS cells (Supplementary Fig. 4f–j). In addition, we found that several of the fatty acid transporters that are responsible for cellular uptake of oleic acid (i.e. ACSL3, SLC27A1, SLC27A2, SLC27A3, and SLC27A5) are significantly upregulated in HCC1954-27HCR cells compared to -27HCS cells (Supplementary Data 4). It has been demonstrated recently that triacylglycerols with a high oleic acid content protect cancer cells from ferroptosis; a finding that is consistent with our data[43]. From these data, we concluded that increased lipid uptake, and to a lesser extent lipid biosynthesis, is a feature of cellular resistance to the antiproliferative actions of 27HC.

Whereas the association between increased lipid uptake and/or biosynthesis was determined to be a hallmark of resistance to 27HC, the extent to which this activity was causally associated with resistance and/or if it contributed to the alterations in cancer cell biology noted was unclear. To address this issue, we repeated the studies probing the sensitivity/resistance of 27HCS and 27HCR variants of 4T1, Py230, HCC1954, B16F10, and BPD6 cells to 27HC in lipid-rich and delipidated media. As shown in Fig. 3c (and in Supplementary Fig. 5a), except for BPD6 cells, resistance to the antiproliferative activities of 27HC was only manifest in lipid replete conditions. Given that the expression of genes involved in cholesterol and fatty acid biosynthesis are upregulated in BPD6-27HCR compared to BPD6-27HCS (see Supplementary Fig. 4a and Supplementary Data 2), it is likely that some of the accumulated lipid in these cells is produced endogenously. The importance of lipid accumulation in the resistant cells was also highlighted in studies showing that lipid depletion reversed the increased in vitro migratory capability of 27HCR cells (Fig. 3d and Supplementary Fig. 5b). Using BODIPY FLC16 staining, we confirmed increased lipid uptake by the 27HCR derivatives of HCC1954, MDAMB436, B16F10, and BPD6. The changes in total lipid content noted were less significant in 4T1 and Py230 cells although we cannot rule out the possibility that specific lipids, important for the resistance phenotype, are increased in these models (Supplementary Fig. 5c). Treatment of 27HCS- and 27HCR-4T1 and Py230 cells with etomoxir (an irreversible inhibitor of carnitine

palmitoyltransferase; an enzyme required for β-oxidation of fatty acids) did not alter total lipid content suggesting that changes in fatty acid oxidation are not responsible for the phenotype of increased lipid accumulation in these 27HCR cells (Supplementary Fig. 5d). Taken together these data indicate that in acquiring resistance to 27HC, the 27HCR cells examined increase their lipid uptake and this activity likely contributes to their increased malignancy.

To define the mechanisms underlying the increased capacity of 27HCR cells to accumulate lipids, we assessed the expression of the primary lipid transporters in different models of 27HC resistance. Among the panel of lipid uptake/transport genes assayed it was determined that the expression of Fabp4 and Vldlr mRNAs are dramatically elevated in 27HCR cells derived from mouse cells (4T1 and Py230), whereas CD36 was found to be overexpressed in the 27HCR derivative of the human HCC1954 line (Fig. 3e and Supplementary Data 4). These findings are interesting in light of published data indicating that the elevated expression of CD36, VLDLR and FABP4 are associated with aggressive cancers and metastasis phenotypes and this has been correlated with increased uptake of exogenous lipids from the tumor microenvironment[44–49]. However, using comprehensive genetic and/or pharmacological approaches, we could not ascribe the resistance phenotype in these cells to a single transporter, a likely consequence of their redundancy. It appears, therefore, that although increased lipid uptake is integrally involved in the resistance phenotype, cells can utilize different mechanisms to facilitate uptake and accumulation of lipids.

**27HC resistant cells are protected from ferroptotic cell death.** Dysregulated lipid metabolism is involved in many aspects of cancer cell pathobiology, from promoting epithelial to mesenchymal transition (EMT) to facilitating resistance to chemotherapy, targeted therapies, and radiation[50–52]. However, the lipid accumulation that we have observed in the 27HC resistant cells is also expected to put considerable metabolic stress on cells as they have to overcome the liabilities associated with the accumulation of lipid peroxides[53]. Recently, it has been shown that cells can tolerate lipid-dependent stress by engaging GPX4 (phospholipid glutathione peroxidase 4)-dependent antioxidant pathways and/or downregulating the expression of the enzymes responsible for lipid oxidation. Thus, we considered that in evolving to a state of 27HC resistance cancer cells may increase the threshold for induction of ferroptosis (increased GPX4 expression or activity and/or develop resistance to ferroptosis by downregulating an essential step(s) in the pathways that generate lipid peroxides). As a starting place for these studies, we assessed the expression of GPX4 in several cell lines (B16F10, Py230, HCC1954, and MDAMB436). Using western immunoblot analysis, it was determined that GPX4 was expressed in all cells and that its expression was downregulated upon treating these cells with increasing concentrations of 27HC (Fig. 4a). This was an important, although not totally unexpected finding, as isopentyl pyrophosphate (from the mevalonate pathway) is needed for the maturation of a very specific selenocysteine-tRNA required for GPX4 synthesis[54]. Indeed, we were able to show that 27HC-dependent downregulation of GPX4 expression was substantially attenuated upon supplementation of cells with exogenous mevalonate (Fig. 4b). It was not obvious how, following GPX4 downregulation, cells could withstand the stress of accumulated lipids as they progress from a 27HC sensitive to a resistant state. To address this question, we probed GPX4 biology in the sensitive and resistant cells and how it was modulated by 27HC. For comparative purposes, we also included as a positive control, erastin, which has been shown previously to downregulate GPX4

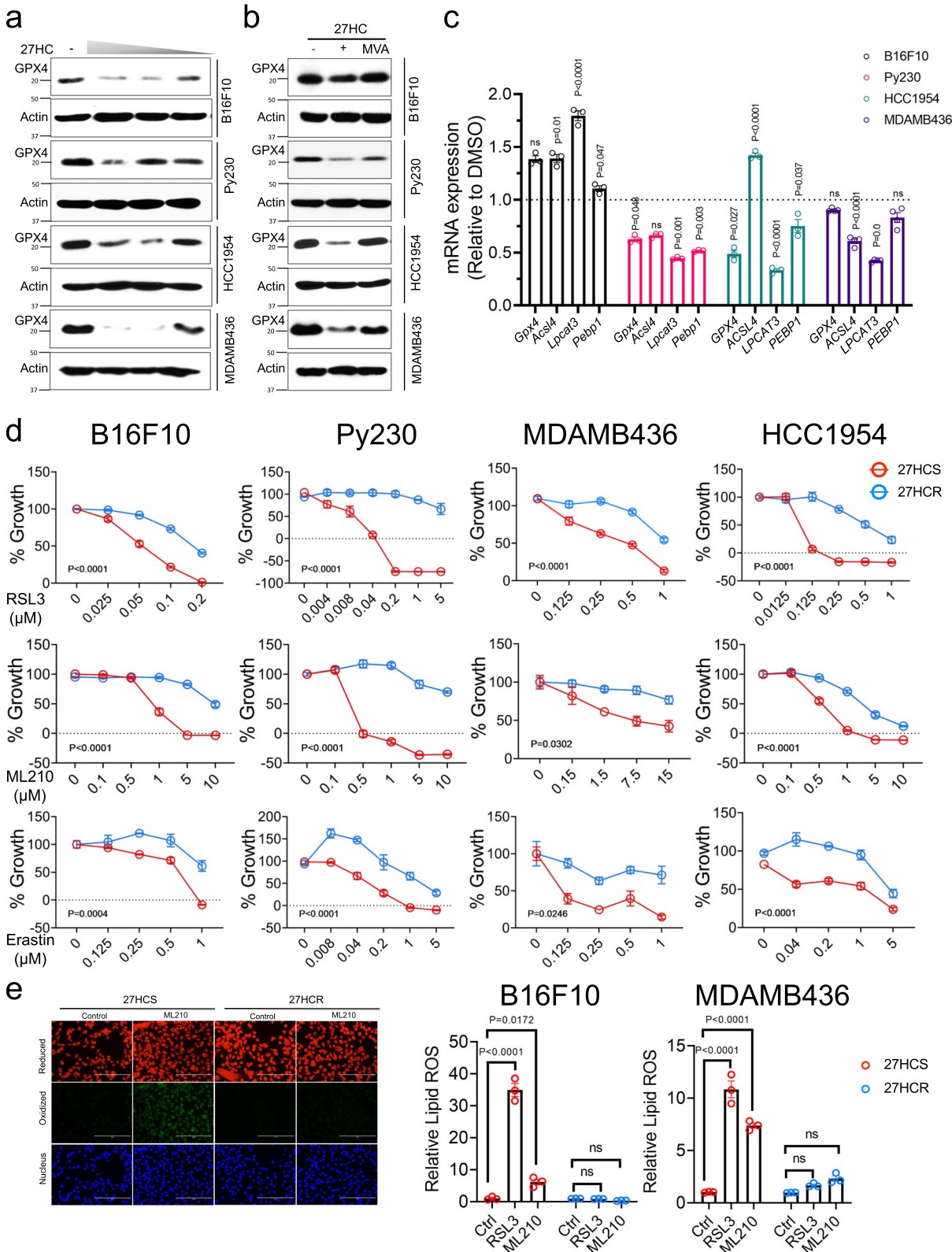

expression[55]. This drug inhibits the Xc-transporter (xCT) and decreases the ability of cells to import cystine an amino acid, which is required for GPX4 activity. It was observed that both 27HC and erastin inhibited the expression of GPX4 in 27HCS cells, albeit to different degrees (Supplementary Fig. 6a). In some 27HCR derivatives (B16F10 and 4T1) a modest increase in GPX4 expression was observed. Likely more important, however, was

the observation that 27HC (or erastin) dependent downregulation of GPX4 expression did not occur in 27HCR cells (Supplementary Fig. 6a). These data indicate that the regulatory mechanisms that enable 27HC (and erastin) dependent downregulation of GPX4 expression are lost in 27HCR cells and this is likely to confer upon these cells a selective advantage. Whereas direct engagement of GPX4 is oneway cells can accommodate lipid

**Fig. 4 27HCR cells are resistant to ferroptosis. a** B16F10, Py230, HCC1954, and MDAMB436 cells were treated with various doses of 27HC for 48 hr. Cell lysates were harvested, and western immunoblots were used to analyze GPX4 expression levels. β-Actin was used as a loading control. Representative results from two independent experiments are shown. **b** mevalonate (MVA, 500 μM) supplementation reversed 27HC-dependent inhibition of GPX4 protein expression in B16F10, Py230, MDAMB436, and HCC1954 cells. Representative results from three independent experiments are shown. **c** qRT-PCR analysis of the expression of genes involved in lipid peroxidation (GPX4, Acsl4, Lpcat3, and Pebp1) in 27HCS and 27HCR cells. Data plotted as mean ± SEM ($n = 3$ wells of cells for B16F10, Py230, and HCC1954; $n = 4$ wells of cells for MDAMB436). $P$ values were calculated using two-sided unpaired Student's $t$ test. **d** 27HCS and 27HCR derivatives of B16F10, Py230, HCC1954, and MDAMB436 cells were treated with inducers of ferroptosis, including GPX4 inhibitors, RSL3 (upper), and ML210 (middle), and the x$_c$- inhibitor, erastin (lower) and 0.1% DMSO control. Morphology of ferroptotic cell death such as cells rounding up and plasma membrane rupture were observed in treated cells. Cell growth was assessed after 48–72 h by measuring DNA content using Hoechst 33258. Relative cell growth (% of Growth) was then calculated by normalizing each treatment condition to control-treated wells. Data plotted as mean ± SEM as representative results from four independent experiments ($n = 5$ wells of cells for B16F10, MDAMB436, and HCC1954; $n = 3$ wells of cells for Py230). $P$ values were calculated using two-way ANOVA. **e** RSL3 (1 μM) and ML210 (5 μM) induced lipid peroxidation in 27HCS and 27HCR-detivatives of B16F10 and -MDAMB436 cells as measured by BODIPY-C11 staining. Fluorescent images of BODIPY-C11 stained B16F10 cells treated with ML210 for 4 h are shown in the left panels. Data plotted as mean ± SEM; $n = 3$ wells of cells. $P$ values were calculated using one-way ANOVA. Figure 4e results were repeated in Py230 cells. Unprocessed immunoblots and numerical source data are reported in the Source Data File.

stress, they can also be protected from this activity by reducing the production of lipid peroxides (lowering the requirement for GPX4). Indeed, in 27HCR derivatives of Py230, HCC1954, and MDAMB436 cells, we observed a quantitative downregulation of the expression of genes the activity of which is associated with increased lipid peroxidation (*Acsl4*, *Lpcat3*, and *Pebp1*) (Fig. 4c). We conclude that the disruption of the regulatory mechanisms that lead to downregulation of GPX stabilizes the expression level of this protein. This activity, together with decreased stress on the GPX4 pathway through modulation of the expression of genes associated with lipid peroxidation (responsible for the production of polyunsaturated fatty acids), is a characteristic of 27HCR cells.

It is important to note that in addition to alterations in the GPX4 pathway, sensitivity to ferroptosis can also be influenced by NRF2-regulated antioxidant defense mechanisms and by alterations in cellular iron metabolism. In this regard, it is important to note that we observed that the xCT genes (*Slc7a11* and/or *Slc3a2*) are upregulated in 27HCR derivatives of B16F10, Py230, and 4T1 and that several mRNAs encoding proteins involved in iron metabolism genes are downregulated in 27HCR-HCC1954 and -MDAMB436 cells (Fig. 4c, Supplementary Fig. 6b and Supplementary Data 5). Regardless of the mechanism, we reasoned that if stabilized expression and/or activity of GPX4 (secondary to a reduction in lipid stress) was causally involved in the 27HC resistance phenotype (and ferroptosis) that 27HCR cells should be more resistant to inhibitors of the GPX4 pathway. As shown in Fig. 4d (and Supplementary Fig. 6c), we observed that B16F10, Py230, MDAMB436, HCC1954, and 4T1 cells are all substantially more resistant to the direct covalent inhibitors of GPX4 (RSL3 and ML210) and the Xc-transporter (erastin). The effects of these drugs on the viability of all of the 27HCS cells examined were reversed by ferrostatin, an inhibitor of ferroptosis, confirming that this is indeed the specific process targeted by these agents (Supplementary Fig. 6d). The importance of this observation was further validated in studies demonstrating that inhibitors of GPX4 (RSL3 and ML210) resulted in a substantial increase in lipid peroxides in 27HCS cells, an activity that was not observed in the 27HCR derivatives of these cell lines (Fig. 4e). We conclude that in adapting to lipid-induced metabolic stress, secondary to the development of resistance to 27HC, cells can avoid ferroptotic cell death by either stabilizing the expression of GPX4, increasing the activity of the GPX4-dependent antioxidant system, or decreasing the production of lipid peroxides.

**GPX4 activity protects against ferroptosis and promotes metastasis.** We considered it possible that resistance to ferroptosis may enable cells to withstand the metabolic and environmental stresses they encounter during the metastatic process. Indeed it has been reported by others in multiple cancer types that increased capacity to withstand oxidative stress increases the metastatic potential of cells[56–59]. We hypothesized, therefore, that resistance to ferroptosis may be a general characteristic of metastatic cells and that chronic exposure to 27HC enriches for cells with this property. To address this possibility, we injected luciferase labeled Py230-27HCS and -27HCR cells intravenously into nude or C57/BL6 mice. After 2 months, Py230-27HCS cells developed micrometastasis, whereas Py230-27HCR cells formed macrometastasis as assessed by luminescence imaging (Fig. 5a, left panels). We harvested cells from these metastatic lesions and evaluated their sensitivity in vitro to inducers of ferroptosis (RSL3 and ML210). The striking result of this study was that metastatic cells isolated from lesions formed by either 27HCS or 27HCR cells were equivalently resistant to these drugs (Fig. 5a, right panels). We conclude that the increased metastatic phenotypes of 27HCR cells are likely a consequence of adaptive events that facilitate 27HC resistance as opposed to a specific change in gene expression/biochemical activity related to treatment with the oxysterol.

Recently, it has been demonstrated that GPX4 has an essential role in protecting against ferroptosis, and this, together with our data indicating that resistance to ferroptosis is a feature of metastatic cells, prompted us to evaluate the impact of disrupting GPX4 expression on the formation of metastases in the highly metastatic B16F10 model (and its 27HCR derivative) and the poorly metastatic Py230 cell (and its highly metastatic 27HCR derivative). For this study, we used an shRNA-based approach to downregulate the expression of GPX4 in B16F10 and Py230 cells (both 27HCS and 27HCR derivatives). The successful knockdown of GPX4 was confirmed by western immunoblot (Supplementary Fig. 7a). GPX4 knockdown slightly decreased the proliferation of B16F10 cells but had minimal effects on the growth of Py230 cells in vitro (Supplementary Fig. 7b). Importantly, we observed that partial knockdown of GPX4 reversed the resistance to GPX4 inhibitors noted above, confirming its essential role in conferring resistance to these drugs (Supplementary Fig. 7c). Next, we probed the specific importance of GPX4 in metastasis. To this end, control cells (GPX4-shScramble control) and two independent cell lines in which GPX4 knockdown had been accomplished in the 27HCS and 27HCR derivatives of B16F10 and Py230 cells, were injected intravenously into C57/BL6 mice. Because the B16F10 cells are so highly metastatic, we were unable to use lesion count as a means to assess differences in metastasis. Therefore, we developed a grading system using a scale of I-IV (Grade I: <10 foci; Grade II:10-50 foci; Grade III: 50-150 foci; Grade IV: countless foci). Using this scale to quantitate metastasis, we

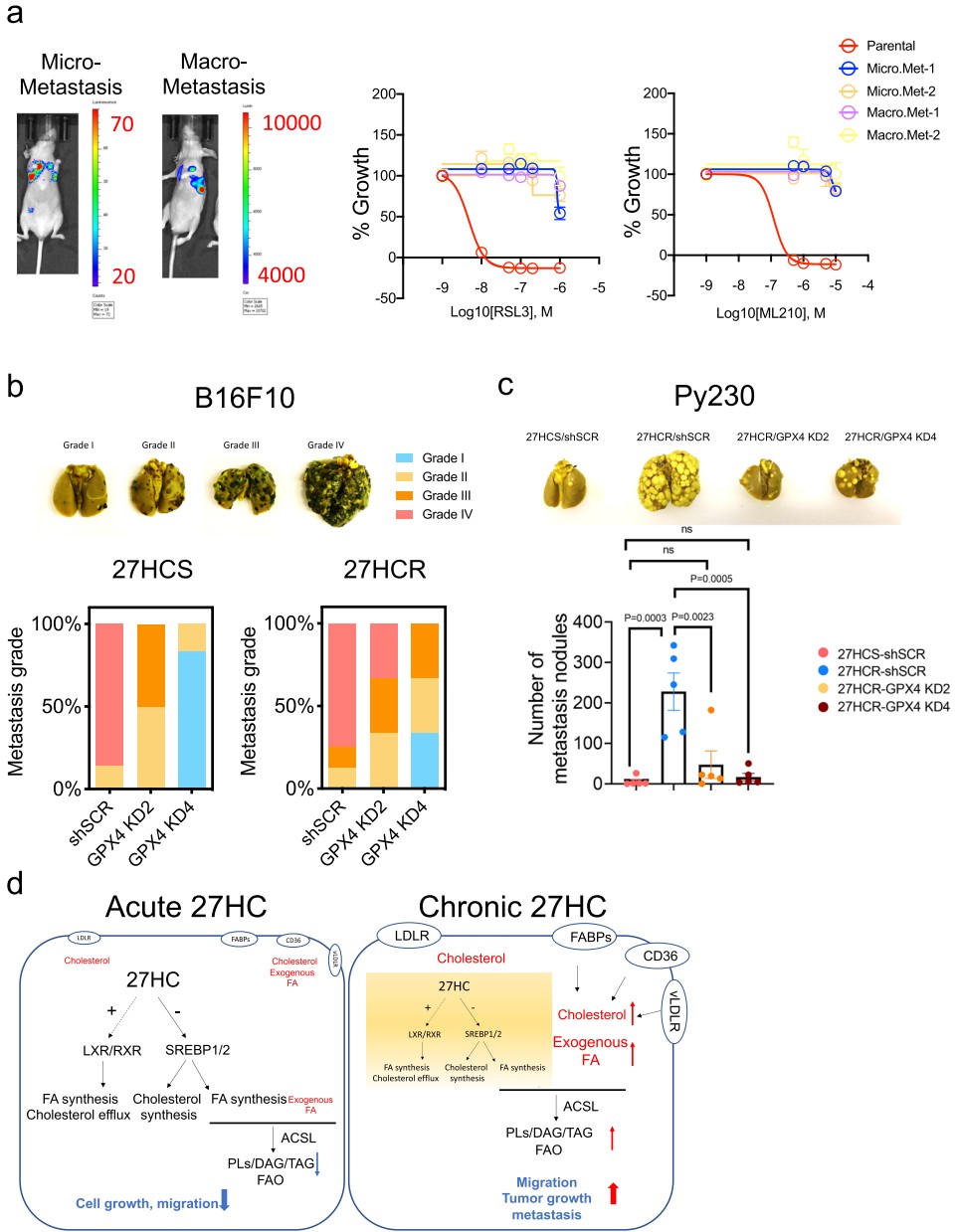

**Fig. 5 GPX4 inhibition sensitizes 27HCR cells to ferroptosis and reduces their metastatic capability. a** A total of $2 \times 10^5$ luciferase labeled Py230-27HCS and -27HCR cells were injected intravenously into nude or C57/BL6 mice. After 2 months, Py230-27HCS cells developed micrometastasis and Py230-27HCR cells formed overt macrometastasis. Bioluminescent images of lung metastasis as shown in the left panels. Metastatic tumor cells were then recovered from lungs and propagated as cell lines (Micrometastasis: Micro. Met1 and 2 and Macrometastasis: Macro Met1 and 2). Py230 parental and lung metastasis derivative cell lines were treated with GPX4 inhibitors, RSL3 (middle), and ML210 (right) for 48 h, followed by cell proliferation assay. Relative cell growth (% of Growth) was calculated by normalizing each treatment condition to control-treated wells. Data plotted as mean ± SEM from five technical replicates. **b** $2 \times 10^5$ GPX4 wild-type (shSCR) and knockdown (KD2 and KD4) B16F10-27HCS and -27HCR cells were injected intravenously into C57BL/6 mice ($n = 5$ mice). Lung metastases were graded and scored when mice were euthanized on Day 16. Representative images of lung tissue used for grading are shown in the upper panel. **c** $3 \times 10^5$ GPX4 wild-type (shSCR) and knockdown (KD2 and KD4) Py230-27HCR cells, as well as GPX4 wild-type (shSCR) Py230-27HCS cells were injected intravenously into C57BL/6 mice ($n = 5$ mice). Lung metastases were quantified when mice were euthanized on Day 40. Representative lung images are shown in the upper inset. Data plotted as mean ± SEM; $P$ values were calculated using two-way ANOVA. **d** The proposed model to explain how cancer cells respond to 27HC treatment. Acute (left) treatment with 27HC disrupts lipid metabolism via interfering with SREBPs and LXR signaling, and this results in the inhibition of cell growth and migration. Cancer cells can adapt to the metabolic stress imposed by chronic treatment by 27HC (right). The cells that survive (27HC resistant cells) increase lipid uptake and accommodate the metabolic stress associated with this activity by upregulating the activity of processes that allow them to withstand lipid oxidative stress (ferroptosis); an activity which confers upon them enhanced tumor growth and metastatic capabilities. Numerical source data are reported in the Source Data File.

demonstrated that knockdown of GPX4 dramatically decreased metastasis in both the 27HCS and 27HCR derivatives of B16F10 cells, albeit to a lesser degree in the latter (Fig. 5b). Likewise, we demonstrated, using direct counting of metastatic nodules, that the increased metastatic phenotype observed in 27HCR derivatives of Py230 cells was abolished entirely upon knockdown of GPX4 (Fig. 5c). These data establish an essential role for GPX4 in preventing ferroptosis during metastatic progression. They also highlight how, in dealing with the stress associated with dysregulated lipid metabolism, cells upregulate the activity of the GPX4 axis and/or reduce reliance on GPX4 to accommodate this stress and this has the collateral liability of increasing the metastatic capacity of cells (Fig. 5d). Interestingly, we have observed that knockdown of GPX4 inhibits the primary tumor growth only in 27HCR—but not 27HCS-B16F10 model (Supplementary Fig. 7d), suggesting that the metabolic reprogramming of tumor cells by chronic exposure to 27HC exposes a reliance on GPX4 as a vulnerability in both the primary tumors and in metastatic lesions.

## Discussion

Dyslipidemia is a common sequela of the overfed state, a comorbidity of obesity and the metabolic syndrome and has been causally linked to the pathobiology of cardiovascular disease and a variety of cancers[60]. In cardiovascular disease, it is clear that elevated cholesterol, cholesterol derivatives, and elevated triglycerides impact cardiovascular disease through a general increase in systemic inflammation and to the specific formation of atherosclerotic lesions within vessel walls[61]. However, the mechanisms by which dyslipidemia impacts cancer pathobiology appear to be more complex and varied. A dominant hypothesis is that the hyperlipidemia associated with overnutrition results in increased pancreatic insulin release and/or increased production of insulin-like growth factor 1 (IGF-1) in the liver[62–65]. Both of these peptide hormones have been shown to function as mitogens and increase tumor growth and metastasis in animal models of several different cancers, providing the rationale for the development of IGFR inhibitors as cancer therapeutics[66,67]. Increased adiposity, secondary to overnutrition and obesity, also contributes to cancer cell growth and metastasis, as adipocytes produce tumor-promoting inflammatory cytokines and express CYP19 (aromatase), allowing for the generation of estrogens from androgens[68]. The latter likely explains the increased risk of estrogen receptor (ER) positive breast cancers, and other ER-expressing gynecological cancers observed in obese women. Indeed the success of aromatase inhibitors and ER-antagonists in treating ER-positive breast cancer in postmenopausal women can be attributed to their ability to inhibit the production or activity of extragonadal produced estrogens[69]. In previously published work, we described a more direct link between obesity/dyslipidemia and breast cancer risk in studies, which demonstrated that 27HC, an oxysterol produced in a stoichiometric manner from cholesterol, functions as a *bona fide* endogenous estrogen and, as such, promotes the growth of ER-dependent cancers[23]. This finding has led to clinical studies to explore the impact of lowering 27HC on the efficacy of endocrine therapy in breast cancer[70].

Notwithstanding the observation that cholesterol, subsequent to its conversion to 27HC, impacts the growth of ER-positive cancers, there remained the question as to how hypercholesterolemia impacted the growth and progression of ER-negative cancers independent of their cell/organ of origin. It was significant, therefore, that through direct actions on neutrophils and indirect actions on γδ T-cells, cholesterol/27HC was shown to prime the metastatic niche, such as to increase cancer cell

metastasis[25]. However, given that 27HC is generally thought to be a negative regulator of cholesterol synthesis and uptake through its ability to inhibit SCAP-dependent SREBP activation[34], and considering the requirement of proliferating cells for cholesterol, it was unclear how hypercholesterolemia could have a net overall positive effect on cancer cell growth and metastasis. The results of the studies herein provide an explanation for this paradox and can be reconciled by taking into account the way we and others have modeled cholesterol biology in cancer using in vitro models. Notably, we demonstrate, in line with what would be expected for a primary cholesterol metabolite, that acute exposure to 27HC inhibits the growth of cancer cells (breast cancer and melanoma were studied) consistent with its known ER-independent pharmacology. However, more chronic treatment of cancer cells with levels of 27HC that reflect those found in hypercholesterolemic patients selected for cells whose growth was resistant to inhibition by 27HC. This can be explained by the ability of cells to both increase their uptake of lipids from the media and in some cases to increased lipid synthesis. This introduced the interesting question as to how cancer cells can bypass the negative feedback on cholesterol/lipid synthesis mediated by 27HC, and increase their uptake of lipids and cholesterol. Whereas we determined that increased lipid content, primarily through increased lipid uptake, was a characteristic of all 27HCR cells, we were unable to define a single pathway that explained this activity. Instead it appeared that cells upregulated the expression of several functionally redundant lipid transporters (VLDLR, FABP4, CD36). The demonstration that the antiproliferative activity of 27HC, at least in the acute setting, could be reversed by cholesterol supplementation alone, suggested that the overall increase in neutral lipid accumulation by 27HC treated cells is a secondary consequence of efforts to specifically replace cholesterol. Interestingly, many cancers rely on exogenous lipids for survival (i.e., Ras-driven cancers rely on lipid uptake[71]) and EGFR mutated glioblastoma requires exogenous cholesterol[72,73]. Thus, the pathways used by cells to bypass the inhibitory activity of 27HC may be used broadly by cancer cells.

Others have shown that mesenchymal cell-derived cancers, or cancer cells exhibiting the characteristics of mesenchymal cells, are highly susceptible to ferroptosis and sensitive to drugs that inhibit the activity or expression of GPX4[74]. This has been attributed to the ability of these cancer cells to increase the synthesis and uptake of polyunsaturated fatty acids (PUFAs) and their resultant transformation into cytotoxic lipid peroxides (and lipid radicals) by the ACSL4/LPCAT3/ALOX pathway. The ability of cells to survive this insult requires that the lipid peroxides are reduced to their alcohol forms by the action of GPX4. Interestingly, the resistance of several epithelial cancers to targeted therapies, manifests as a shift towards a mesenchymal phenotype, is associated with increased lipid uptake, the formation of lipid peroxides, and increased sensitivity to inhibitors of GPX4[74–76]. Forced expression of the mesenchymal transcription factor ZEB1 in epithelial cells recapitulates these activities, establishing a cause and effect relationship between mesenchymal phenotype, lipid uptake, lipid toxicity, and the requirement for GPX4[74]. Of particular importance to this study is the observation that the stress on the GPX4 system can be reduced by genetic knockdown of the enzymes responsible for the production of lipid peroxides from PUFAs (ACSL4, LPCAT3, and ALOX), and therefore increased resistance to GPX4 inhibitors[74]. We have observed that all of the 27HCR cells we created are more resistant to GPX4 inhibitors and to different degrees, this can be explained by stabilization of the expression of GPX4, increased activity of GPX4 and/or decreased ferroptotic stress by downregulation of expression of ACSL4 and LPCAT3, which are required for the biosynthesis of polyunsaturated phospholipids and their

subsequent peroxidation. We also observed upregulation of xCT transporter genes (Slc7a11 and Slc3a2) and decreased expression of iron metabolism genes, the significance of which is currently under investigation. Regardless, the knockdown of GPX4 in 27HCR cells restored sensitivity to GPX4 inhibitors to a level that resembles their isogenic 27HCS counterparts. These results highlight the primacy of GPX4 in adaptive responses to lipid accumulation that are noted in cells that are resistant to 27HC.

One of the most significant findings in this study was the demonstration that the increased activity (or decreased stress) of the GPX4 pathway was causally linked to the enhanced metastatic activity of 27HCR cells. We demonstrated that the genetic knockdown of GPX4 reduced the metastatic capacity of 27HCR B16F10 melanoma cells, when compared to that of their wild-type counterparts. Further, the gained metastatic activity in the 27HCR Py230 model was entirely lost upon GPX4 knockdown. Metastatic dissemination requires cells to detach from the extracellular matrix (ECM) an activity that has been shown to result in the generation of intracellular ROS, which inhibits fatty acid oxidation (FAO)[77,78]. Cells respond to this activity by increasing their uptake of exogenous lipids, which, as has been discussed above, puts cells at risk of ferroptotic cell death[46,79,80]. Dietary supplementation of antioxidants or genetic manipulation of cells to increase glutathione synthesis (will also increase GPX4 activity) has been shown to promote distant metastasis in animal models of lung cancer and melanoma[56–59,81]. Whereas these previous studies did not make the link to GPX4 or ferroptosis they are consistent with the findings of this study. Initially, we considered that the increased GPX4 activity and metastatic capacity in the 27HCR cells was specifically related to treatment-specific effects of 27HC. However, the observation that the basal level of metastasis in the treatment naïve 27HCS B16F10 cells was also reduced upon GPX4 knockdown suggested that this enzyme/pathway, and resistance to ferroptosis, may contribute in a fundamental way to metastasis. Recent studies have highlighted the importance of antioxidant defense mechanisms to protect metastasizing cancer cells from ferroptotic cell death while in circulation and within the metastatic niche[43,82]. This is in agreement with our observation that cells harvested from the lung metastasis of either parental or 27HCR cells were equally resistant to inhibitors of the GPX4 pathway. This suggests that rather than specifically reprogramming cells to make them more metastatic per se, 27HC selects for a subpopulation of cells that exhibit increased GPX4 activity/reduced production of lipid peroxides and it is this property that increases their propensity to metastasize.

The therapeutic implications of this work are significant in that they highlight several points of intervention that can be exploited to decrease the impact of hypercholesterolemia and dyslipidemia on tumorigenesis and tumor progression and also demonstrate the importance of targeting the GPX4 axis in cancers, in general, as a means to treat metastatic disease. The data suggesting that statins reduce breast cancer risk are equivocal, possibly due to differences in the mechanism of action of different statins and to the considerable size of trials needed to see robust positive effects in the primary prevention setting. There are data demonstrating that overall survival is improved in most cancers in patients on a statin at the time of diagnosis suggesting that this drug may be modulating some aspect(s) of cancer cell biology[8–13]. However, the use of statins in the preventative or treatment setting needs to be explored further. Whereas we have been able to show that statins (atorvastatin) decrease circulating 27HC levels in breast cancer patients it was also demonstrated that within the tumors of treated patients there was a compensatory increase in the expression of HMG-CoA reductase, CYP27A1, and decreased expression of reverse cholesterol transporters[26]. Thus, statin-treated cancer cells increase their uptake of exogenous cholesterol (and likely also lipids) or increase intracellular cholesterol biosynthesis, a result also observed in vitro[83]. As with 27HC, the former activity would increase the need for increased GPX4 activity, and the latter would increase the need for production of the prenylation products needed for GPX4 synthesis. It remains to be determined whether long-term statin treatment will enrich for a population of resistant cells similar to what we have found by placing cells under the pressure of 27HC. These data, however, suggest that the treatment of cells with CYP27A1 inhibitors, blocking the conversion of cholesterol to 27HC, may have a more specific impact on cancer cell biology. We have shown in vivo that CYP27A1 inhibitors decrease the growth of ER-positive breast tumors, but the findings in this study suggest that they may have broader utility as cancer treatments, especially in the background of dyslipidemia.

Whereas the primary objective of this work was to understand how dyslipidemia impacts breast cancer pathology, we also identified a central role for GPX4, and resistance to ferroptosis, in metastasis. GPX4 and the GPX4 pathway have received considerable attention of late as therapeutic targets, especially in mesenchymal tumors, those that become resistant to targeted therapies and exhibit mesenchymal characteristics and in neuroendocrine tumors[76]. Most studies of these drugs have looked at the activity of GPX4/ferroptosis inhibitors in vitro, or in some cases on primary tumor growth. However, the central role of this pathway in metastasis was unexpected raising the possibility that such interventions may have use in the treatment of advanced disease. There are several GPX4 inhibitors in development although it remains to be determined if their therapeutic index will be good enough to permit their use in cancer patients[84].

## Methods

**Reagents**. 27HC was synthesized by Sai Life (Hyderabad, India). LXR agonist GW3965, and antagonist GSK2033, Cholesterol, 2-Hydroxypropyl-β-cyclodextrin, Mevalonolactone (MVA), Isopentenyl pyrophosphate triammonium salt solution (IPP), Farnesyl pyrophosphate ammonium salt (FPP), Geranylgeranyl pyrophosphate ammonium salt (GGPP) were obtained from Sigma. Cholesterol was solubilized in 40% (2-hydroxypropyl)- β-cyclodextrin at room temperature, sterile filtered (0.45 μM), and stored at −20 °C. MVA was resolubilized with 0.1 M NaOH, followed by neutralization with 0.1 M HCL/1 M HEPES. Triacsin C (ACSL4 inhibitor), and BMS 309403 (Fabp4 inhibitor) were obtained from Tocris. (1 S,3 R)-RSL3, ML210, Erastin, Ferrostatin-1, and Sulfosuccinimidyl Oleate (SSO) were purchased from Cayman Chemicals. Delipidated serum was purchased from Cocalico Biologicals (Stevens PA).

**Cell lines and culture**. 4T1, Met1, E0771, MDAMB436, BPD6, B16F10, 293FT cells were cultured in DMEM media (Fisher Scientific); HCC1954 cells were cultured in RPMI-1640 media (Fisher Scientific). MCF7 cells were maintained in DMEM/F12 media. DMEM, DMEM/F12, and RPMI media were supplemented with 8% fetal bovine serum (FBS, Gibco), 1 mM sodium pyruvate, and 0.1 mM nonessential amino acids (Invitrogen). Py230 and Py8119 cells were cultured in F12K media (Fisher Scientific) supplemented 5% FBS and Mito + serum extender (Fisher Scientific). All cells were cultured in a humidified incubator at 37 °C with 5% CO₂. For experiments with MCF7 cells, cells were plated in the same media lacking phenol red and supplemented with 8% charcoal-stripped FBS (stripped twice, 2XCFS). For experiments with delipidated serum-containing media, cells were starved with serum-free media for 24 h then proceeding treatments with delipidated serum-containing media. HCC1954 cells were luciferase and GFP labeled, BPD6 and B16F10 cells and 293FT cells were kindly provided by Dr. Joan Massague (Memorial Sloan Kettering Cancer Center), Dr. Brent Hanks (Duke University), and Dr. Kris Wood (Duke University), respectively. Py230 and Py8119 cell lines were kindly provided by Dr. Lesley Ellies (University of California- San Diego). All other cancer cell lines were obtained from American Type Culture Collection (ATCC). Py230 and Py8119 cells were transduced with lentivirus containing fusion protein reporter construct encoding firefly luciferase and green fluorescent protein (GFP) and made stable cell lines. All cell lines were tested using a PCR-based assay and found to be negative for mycoplasma.

**Generation of 27HC resistant cell lines**. Two sets of 4T1, MDAMB436, HCC1954, Py230, B16F10, and BPD6 cells were seeded into 10-cm dishes. 27HC (5 μM) and 0.1% DMSO (vehicle) were added to media the next day and

maintained (changed every 2–3 days) during selection (1–4 months) and 2 weeks post selection. For 4T1, HCC1954, B16F10, and BPD6 cells, 27HC killed most cells in the first week; for MDAMB436 and Py230 cells, 27HC did not induce massive cell death but inhibited cell proliferation. 27HC-tolerant cells that evolved from 4T1, BPD6, and Py230 models were isolated, expanded to one confluent 10-cm dish, and frozen. In HCC1954, B16F10, MDAMB436 models, 27HC-tolerant colonies (10~50) were isolated and transferred to 12-well plates. The few colonies that survived 27HC treatment in 12-well plates were expanded to 6-well plates and 10-cm dishes before freezing. For 27HC resistant 4T1 cells selected in 3D culture (4T1-RSA and 4T1-SSA), 5000 4T1 cells were seeded in soft agar and cultured in the presence of 1 μM 27HC or 0.1% DMSO for 3 weeks. The 27HC-tolerant colonies (and DMSO-treated colonies) that emerged were isolated and transferred into 12-well plates and maintained in 27HC and DMSO treatment which were replaced every 3 days. Confluent cells were expanded in 10-cm dishes and subjected to second and third rounds of 27HC selection in soft agar.

**Cell proliferation assay**. Three thousand to 5000 cells were seeded in 96-well plates containing regular full-serum media and allowed to adhere overnight. Treatments were added the next day (except Py230 cells, treatments were added 2 days after seeding), and cells were harvested at times indicated, and were assayed for DNA content using Hoechst 33258 (Sigma). Fluorescence intensity was read at excitation 346 nM and emission 460 nM using a plate reader (TECAN Spark). Data shown are representative of three independent experiments (raw fluorescence values) (Fig. 1a, c, f, and Fig. 2a) or normalized by seeding density and vehicle control (Fig. 2c) or % of growth (Fig. 4d and Fig. 5a).

**Soft agar assay**. Colony formation was assessed by monitoring cancer cell growth in soft agar. Briefly, 0.6% agar in growth medium (2 ml) was added to a 6-well plate and allowed to solidify. Five thousand cells per well were suspended in 2 ml of 0.3% agar in growth media with treatments and added on top of the base agar and allowed to solidify. Media containing treatments were added to each well and replaced every three days. Colonies were stained with crystal violet and counted using a phase-contrast microscope.

**Migration assay**. Migration of cancer cells was evaluated using wound healing and transwell migration assays. For wound healing assays, cells were seeded in a 12-well plate and cultured to a confluent monolayer, media was then replaced with serum-free (or 0.02% FBS-containing media), and cultured overnight. The next day, two scratches were introduced per well by scraping the monolayer with a p200 pipette tip and marked for orientation. At least six images were photographed by microscopy immediately (T1) and after 24 h (T2). Scratch introduced open area (OA) was measured by Image J and evaluated by Migration Index = [(OAT1 − OAT2)/OAT1] × [OAT1/Ave of all OAT1]. For transwell migration assays, transwell chambers (BD Biosciences) were used as described[85]. Briefly, cells were cultured in serum-starved conditions overnight and then $5–8 \times 10^4$ cells were seeded in the upper chamber of a Transwell chamber (8 μm pore size polycarbonate membrane filter) in 0.3 ml of 0.2% FBS (or DL-FBS)-containing media, and 0.7 ml of 8% FBS (or DL-FBS) containing media in the lower chamber. Cells were incubated 12–24 h. Cells that migrated through the membrane were fixed with 4% paraformaldehyde and stained with 0.1% crystal violet for 15 min at room temperature and photographed under the microscope. A minimum of five images were photographed, and each sample condition was performed with at least three technical replicates. Data shown are representative of three independent experiments.

**Gene expression analysis by qRT-PCR**. RNA was extracted using the Aurum RNA isolation kit (Bio-Rad Laboratories, Hercules, CA) following the manufacturer's instructions. cDNA was synthesized using the Bio-Rad iScript kit. 0.5 μg of total RNA was used in each 20 μl reverse transcription reaction, and the resulting cDNA was diluted 1:10 with water for qPCR analysis. qPCR was performed using the Bio-Rad SYBR green supermix with 0.2 μM of each forward and reverse primer and 2.5 or 1.25 μl of diluted cDNA in a total reaction volume of 6.5 or 3.25 μl. PCR amplification was carried out using the Bio-Rad iQ4 or the CFX384 qPCR system. All primers used in the study were tested to have a PCR efficiency between 100+/−10% and span intron/exon boundaries if possible. Gene expression levels were first normalized to an internal control gene, 36B4, and then to control conditions unless otherwise indicated. qPCR primers are listed in Supplementary Data 6. Each experiment was performed with at least three technical replicates. Data shown are representative of three independent experiments.

**Immunoblot**. Cells were briefly washed with cold PBS, collected and resuspended in lysis buffer (50 mM Tris-Cl pH 7.4, 1% NP40, 0.25% sodium deoxycholate, 150 mM NaCl, 1 mM EDTA) supplemented with 1X protease inhibitor cocktails (Roche), 1 mM $Na_3VO_4$ (Sigma), and 5 mM NaF. Protein concentrations were determined by the BCA protein assay (Thermo Fisher Scientific). The lysates were boiled for 10 min, and 15–20 μg of protein was resolved by SDS-PAGE on an 8% polyacrylamide gel, transferred onto nitrocellulose membrane, and subjected to immunoblotting using GPX4 Rabbit polyclonal antibody (ab41787, 1:1000, Abcam) and Actin Mouse monoclonal antibody (A5441, 1:20,000, Sigma). The membranes

were then incubated with HRP-conjugated secondary antibodies (1:5000, BioRad #1706516 Anti-Mouse IgG, and #1706515 Anti-Rabbit IgG) and bands were visualized by ECL plus system (PerkinElmer).

**Lipid droplet, uptake, and lipid ROS assays**. Lipid droplet content was assessed by staining with standard dye BODIPY 493/503 (Thermo Fisher). 27HCS or 27HCR cells (3000–10000 per well) were seeded in 96-well plates in regular media and grown to 60% confluence followed by BODIPY 493/503 staining following published methods[86]. For the lipid uptake assay, BODIPY FL C16 (Thermo Fisher) was diluted to 5 mM in DMSO as stock solution. 27HCS or 27HCR cells were plated as above. The next day, cells were washed with cold PBS and BODIPY staining solution (2.5–100 μM BODIPY, 5 mM Fatty acid-free BSA in PBS) were added. Cells were then incubated at 37 °C for 25 min. Cells were washed three times with cold PBS and incubated with 4% paraformaldehyde (PFA) for 30 min at RT. Cells were further stained with 1 μg/ml Hoechst 33342 for 30 min at RT. BODIPY and nuclei Hoechst fluorescence were measured by a plate reader (TECAN Spark). For lipid peroxidation analysis, 5000 cells were plated per well in a 96-well plate and cultured for 48 h. Cells were then treated with 0.1% DMSO (vehicle), RSL3 (1 μM), and ML210 (5 μM) for 4 h. The treatment media was removed, and cells were washed once with HBSS, followed by labeling in 100 μl HBSS per well containing 5 uM BODIPY 581/591 C11 (Thermo Fisher) and 5 mM fatty acid-free BSA for 30 min at 37 °C. Cells were washed with PBS and fixed with 4% paraformaldehyde (PFA) for 30 min at RT. Cells were further stained with 1 ug/ml Hoechst 33342 for 30 min at RT. BODIPY and nuclei Hoechst 33342 fluorescence were measured using a plate reader (TECAN Spark).

**Lentivirus Infection**. For knockdown experiments, short hairpin RNAi targeting vLDLR and GPX4 was purchased from Dharmacon (Horizon). Lentiviruses were generated in 293FT packaging cells in 10 cm dishes or 6-well plates using FUGENE6 (Promega), and used to transduce Py230, BPD6, and B16F10 cells. Multiple hairpin constructs were screened for knockdown efficiency. Transduced cells were treated with puromycin for 2–3 weeks before gene-knockdown and evaluated by qRT-PCR or Western blot. For GPX4 knockdown cells, Ferrostatin-1 (5 μM) was added to media after lentivirus transduction and maintained for the selection period. GPX4 knockdown cells were maintained a minimum of 7 days before in vitro and in vivo assays.

**Lipid metabolite extraction**. HCC1954-27HCS and -27HCR cells were seeded into 6-well plates at the density of 100,000 cells per well. After an overnight incubation in full growth RPMI medium, the spent medium was replaced with full growth media (lipid rich) or 2XCFS containing media (lipid poor), incubated for 48 h, grown to 80% confluency and subjected to lipid extraction. Media was aspitated at room temperature, cells were washed with ice-cold PBS, and the plates were immediately placed on dry ice where 1 ml of 80% precooled methanol/water (HPLC grade) were added per well. The plates were then transferred to a −80 °C freezer for 15 min to further inactivate enzyme activities. The plates were kept on dry ice until cells were transferred into individual 15 ml plastic tubes. Cells were then vortexed for 1 minute, 2.4 ml of ice-cold MTBE (HPLC grade, precooled on ice) was added on ice, and vortexed for 1 min. Next, 0.6 ml ice-cold water was added and vortexed for 1 min. The final extraction solvent composition is MeOH/MTBE/H2O1:3:1 (v/v/v, all HPLC grade). The extracts were centrifuged at 20,000 rcf at 4 °C for 10 min, two layers formed, and the top layer containing lipids was transferred into two clean 2 ml Eppendorf tubes. The top layer supernatant was dried in a vacuum concentrator at RT. The dry pellets were reconstituted into 30 μl isopropanol (IPA), vortexed for 1 min, and centrifuged at 3500 rcf at 4 °C. 3ul supernatant was injected and further analyzed by liquid chromatography-mass spectrometry (LC-MS).

**Mass spectrometry**. The QE-MS is equipped with a HESI probe, and the relevant parameters are as listed: heater temperature, 120 °C; sheath gas, 30; auxiliary gas, 10; sweep gas, 3; spray voltage, 4 kV for both the positive mode and negative modes. The capillary temperature was set at 320 °C, and S-lens was 65. A full scan range from 200 to 1500 (m/z) was used. The resolution was set at 70,000. The maximum injection time (max IT) was 200 ms with typical injection times around 50 ms. These settings resulted in a duty cycle of around 550 ms to carry out scans in both the positive and negative modes. Automated gain control (AGC) was targeted at $3 \times 10^6$ ions. Mass calibration was performed before any sample analysis.

**High-performance liquid chromatography**. Ultimate 3000 UHPLC (Dionex) is coupled to Q Exactive Plus-Mass spectrometer (QE-MS, Thermo Scientific) for lipid profiling. A reversed-phase liquid chromatography method (RPLC) employing an Xbridge BEH C18 column (100 × 2.1 mm i.d., 2.5 μm; Waters) at 40 °C is used for lipid separation. The mobile phase A is 80:20 (v/v) water: acetonitrile with 0.1% formic acid and 10 mM ammonium formate. The mobile phase B is 90:10 (v/v) isopropanol:acetonitrile with 0.1% formic acid and 10 mM ammonium formate. The linear gradient used was as follows: 0 min, 40% B; 1.5 min, 40% B, 5 min, 85% B; 12 min, 97% B, 16 min, 97% B, 16.5 min, 40% B, 20.5 min, 40% B. The flow rate was 0.2 mL/min.

**Lipidomics data analysis**. LC-MS peak extraction and integration were performed using commercially available software Sieve 2.2 (Thermo Scientific). The peak area was used to represent the relative abundance of each metabolite in different samples and normalized to cell numbers in each individual sample.

**Animal studies**. All xenograft procedures were approved by the Duke University Institutional Animal Care and Use Committee. Viable 27HCS-, and 27HCR versions of 4T1 (200,000), Py230 (1,000,000), BPD6 (200,000), and HCC1954 cells (3,000,000) (suspended in HBSS/Matrigel, 1:1) were inoculated orthotopically into the mammary fat pads of 8-week-old female Balb/C (4T1), or C57BL/6 (Py230 and BPD6) or athymic nude (HCC1954) mice, respectively. C57BL/6 and Balb/C mice were purchased from Jackson Laboratories and nude mice obtained from the Cancer Center Isolation Facility (Duke Cancer Institute, Durham, NC). Mice were maintained under specific pathogen-free, temperature- and humidity-controlled conditions, with a 12-h light/12-h dark schedule. During tumor studies, mice were monitored daily, and tumors volume was measured as an index of the growth rate and calculated as (width + length)/2 × width × length × 0.5236. Experimental metastasis was studied by injecting viable 27HCS-, and 27HCR of 4T1 (200,000), Py230 (300,000–500,000), BPD6 (200,000) cells, with or without luciferase labeling (suspended in ice-cold 200 ul HBSS) into the lateral tail veins of athymic or syngeneic mice. Mice were euthanized at indicated time points. Organs were then collected, and lungs were inflated with cold PBS, collected, and placed in Bouin's solution (Sigma). Metastatic lesions were counted macroscopically. Statistical analyses were performed by two-way ANOVA followed by a Bonferroni multiple comparison test.

**Statistical analysis**. Experiments were done using a minimum of three replicates for each experimental group. Representative data from at least two replicate experiments are depicted. For all statistical analyses, GraphPad Prism software was used. Statistical significance was determined by a two-sided Student $t$ test, one-way ANOVA followed by Tukey's post hoc test, or two-way ANOVA followed by a Bonferroni multiple comparison test. Statistical significance was defined as $P < 0.05$.

**Reporting Summary**. Further information on research design is available in the Nature Research Reporting Summary linked to this article.

## Data availability

All data supporting the findings of this study are available in the article and in the supplementary information. Source data are provided with this paper.

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

## Acknowledgements
We thank members of the McDonnell lab for insights and discussion, and Dr. Xiaojing Liu (Integrated Metabolomics Resource, Duke University) for the lipid LC-MS analysis. This work was supported by BC151638 (DOD) and DK048807 (NIH).

## Author contributions
W.L., R.S., C.Y.C., and D.P.M. designed the experiments. W.L and B.C. performed and analyzed the majority of the experiments with help from R.S. D.K. performed biostatistical analyses of RNA-seq data. The manuscript was written by W.L. and D.P.M. with input from all of the authors.

## Competing interests
The authors declare no competing interests.
