## [Peer Review File · Nature Communications]

Dysregulated cholesterol homeostasis results in resistance to ferroptosis increasing tumorigenicity and metastasis in cancerREVIEWER COMMENTS

Reviewer #1 (Remarks to the Author); expert on lipid metabolism and cancer:

In this manuscript, the authors show that acute treatment with 27-hydroxycholesterol, 27-HC, an intermediate of cholesterol degradation that is elevated in patients with hypercholesterolemia, causes growth inhibition in ER negative breast cancer and metastatic melanoma cells. This is in contrast to their previous findings in ER-positive breast cancer, where 27-HC acts growth promoting by functioning as a partial ligand of the oestrogen receptor. The authors then go on to generate TNBC cells that are made resistant to the effects of 27-HC. While the acute inhibitory effect of the compound on expression of target genes of SREBP and LXR is still intact, the 27-HC resistant populations show an enhanced capacity for tumour growth and metastasis formation. The authors also observe that resistant cells accumulate higher levels of several lipid classes, most likely due to enhanced lipid uptake. As lipid accumulation has previously been linked to ferroptosis, an iron-dependent form of cell death mediated by lipid peroxidation, the authors next investigated whether 27-HC resistant cells show different activity of the ferroptosis machinery. They found that 27-HC resistant cells show differential expression of several genes linked to lipid peroxidation, namely GPX4, ACSL4, LPCAT3 and PEBP1. They also observed increased resistance towards GPX4 inhibitors (RSL3 and ML210) as well as reduced lipid peroxidation in 27-HC resistant cells. Finally, the authors demonstrate that silencing of GPX4 reduces metastasis formation by B16F10 melanoma and Py230 breast cancer cells. The authors conclude that chronic exposure to 27-HC selects for cells that show an enhanced resistance towards ferroptotic cell death and thus promotes the ability of cancer cells for metastatic outgrowth.

This study contains a large amount of data generated across several cellular systems, as 5 breast and two melanoma cell lines were used for most of the experiments. It describes a very interesting concept of selection during chronic exposure to a compound that is found in the circulation of patients with hypercholesterolemia, thus elucidating the mechanism behind the observed increased cancer risk. While most experiments are very clear and adequately performed and the conclusions are sound, there is some lack of clarity in the underlying concept that should be addressed before publication.

The main point of criticism is that the authors show that 27-HC resistant cells accumulate lipids, most likely due to enhanced uptake, which makes them vulnerable to lipid peroxidation. They then hypothesise that pathways that protect against lipid peroxidation, i.e. GPX4, should be upregulated to support survival in resistant cells. Alternatively, enzymes that promote lipid peroxidation can be downregulated. They therefore investigate the expression of GPX4 and three enzymes that have previously been shown to be required for lipid peroxidation (i.e. ACSL4 and AGPAT3, which are required for the incorporation of poly-unsaturated fatty acids into membrane lipids, and PEBP1 that mediates the access of lipoxygenases to membrane lipids) in the panel of 27-HC sensitive and resistant cells. However, the differential expression of these factors varies widely across the cell lines used. For example, 27HC resistant HCC1954 cells show downregulation of GPX4, while ACSL4 is upregulated. Nevertheless, the cells show a marked resistance towards the GPX4 inhibitors and erastin. In contrast, B16F10 melanoma cells upregulate all of the tested factors but still show decreased sensitivity towards ferroptosis induction.

Given this lack of correlation, it is possible that the altered expression of the investigated factors is not, or not solely, responsible for ferroptosis resistance. Instead, the observed difference in ferroptosis sensitivity could actually be caused by alterations in lipid composition, which has previously been linked to ferroptosis sensitivity. While the authors have compared overall lipid levels between sensitive and resistant cells, they did not specify changes in PUFA-containing lipids. Another possible mechanism could involve the activation of NRF2 and the corresponding induction of processes that protect against oxidative stress, for example through increased production of GSH and NADPH. This possibility could also be investigated. Nevertheless, the authors show very convincingly that depletion of GPX4 reduces the metastatic capacity of the two cancer cell lines studied. In the absence of a

clearer mechanism mediating 27-HC resistance, the results should be more cautiously discussed.

Below are also some more specific issues that should be addressed.

- 1) It should be specified whether the 27-HC treatment shown in Fig 1C and S1A was done in delipidated medium.
- 2) The conclusion that 27-HC resistant cells activate lipid synthesis is not supported by the data. Accumulation of TAGs could also be the consequence of reduced lipid oxidation.
- 3) The increased signal of Bodipy dye to demonstrate enhanced lipid uptake shown in Figure 3E is very small for 4T1 and Py230 cells. Is this difference significant?
- 4) The investigation of different inhibitors towards components of the lipid uptake/remodelling machinery shown in Figure S6 is not very clear.
- 5) Line 329. Erastin is an inhibitor of xCT and blocks GSH synthesis rather than downregulates GPX4. This should be clarified in the text.
- 6) Line 341: More rationale should be provided why only ACSL4, LPCAT3 and Pebp1 were investigated as potential mediators of lipid peroxidation.
- 7) Line 370: A link between ferroptosis resistance and metastasis formation has been described before (Alvarez et al Nature 2017 and Ubellacker et al. Nature 2020). These references could be included in a revised version of the manuscript.
- 8) The display of the qPCR data as heatmaps (Fig. 1E, 3A and 4C) is quite unusual and makes the data hard to interpret. Do the panels represent biological replicates from the same pool of sensitive or resistant cells or independently generated pools/clones of resistant populations? This should be clarified.

Reviewer #2 (Remarks to the Author); expert on ferroptosis and cancer:

In the present manuscript, Liu et al. reports that in contrast to the pro-tumorigenic role of 27-hydroxycholesterol (27-HC) in ER-positive breast cancer cells, 27-HC treatment attenuates the proliferative and migratory potential of ER-negative breast cancer cells. Additionally, these cancer cell line models can acquire resistance to 27-HC under chronic selection. Importantly, the 27-HC-resistant (27HCR) cells appear to exhibit higher tumorigenic and metastasis formation potential than their non-selected, parental ER-negative breast cancer cells. 27HCR cells also display higher levels of lipid accumulation, and decreased sensitivity to GPX4 inhibition induced ferroptosis – both features may be relevant contributors to the increased tumorigenesis and metastasis capacity of the 27HCR cells.

Overall, the results in this work are both interesting and can be important. The findings related to 27-HC resistance are novel and of interest to the cancer research community. The data are largely consistent across multiple cell line models that were assessed in this study, therefore are largely convincing. Together, the work is worth considering for publication at Nature Communications, given that the following concerns are properly addressed:

Major questions:

1. The authors have proposed that the upregulation of lipid receptors are crucial to increased lipid accumulation in 27HCR cells, and showed that knockdown the expression of a single lipid transporter vLDLR is not sufficient to suppress lipid accumulation. Does overexpression of vLDLR, FABP4 or CD36 at least partially recapitulate the lipid accumulation phenotype in the specific ER-negative cancer cells used, and how does the expression of these genes alter the sensitivity to ferroptosis?
2. In this referee's opinion, the logic of the experiments described in the section "27HC resistant cells are protected from ferroptotic cell death" and Figure 4 can be optimized: When there is increased lipid accumulation, a plausible assumption is that cancer cells may experience increased lipid oxidative stress and consequently alterations in the sensitivity to GPX4 inhibition-induced ferroptosis. Lipid accumulation is often insufficient to trigger the speculation that cells would compensatorily downregulate lipid repair enzymes like GPX4 or lipid biosynthesis genes. (Also a minor point, ACSL4

and LPCAT3 are not genes required for lipid peroxidation, they are enzymes required for the biosynthesis of polyunsaturated phospholipids, which are substrates of lipid peroxidation.) (Alternatively, the authors may consider introducing their characterizations of the GPX4 inhibitor sensitivity in the 27HCS and 27HCR cells before their investigation into GPX4 expression levels.)

Related to this issue, in line 336-337, the authors stated that “These data suggest that the synthesis of GPX4 is dysregulated in 27HCR cells in favor of increased expression of GPX4.” This conclusion appears largely unsupported by the data in Figure 4C and Supplementary Figure 7, since the GPX4 expression level in 27HCR does not seem to be higher than that in the 27HCS cells (at least in the absence of 27HC treatment, which is the same condition as the tumor implantation and metastasis assays). The authors may want to clarify this issue.

Again related to the same issue, in line 380-381, the authors stated “that the increased metastatic phenotypes of 27HCR cells are likely a consequence of adaptive events that facilitate resistance (i.e., selection of cells that exhibit increased GPX4 activity)...”. This conclusion appears unsupported by experimental data, since Py230 cells do not seem to exhibit increased GPX4 expression in the 27HCR derivatives (Figure 4C and Supplementary Figure 8A).

In this referee’s opinion and according to the lipidomics result, the increased resistance to ferroptosis in the GPX4 inhibitors in 27HCR cells are likely at least partially caused by the increased accumulation of lipids containing saturated or monounsaturated fatty acids, and the relatively limited accumulation of polyunsaturated phospholipids (true substrates for lipid peroxidation and ferroptosis). Note that increased levels of monounsaturated fatty acids are inhibitory of ferroptosis sensitivity (Magtanong et al., Cell Chemical Biology 2019), which is further demonstrated in a more recent study (Ubellacker et al., Nature 2020).

3. Line 406-407, “These data establish an essential role for GPX4 in preventing ferroptosis, an activity that has heretofore not been linked to metastasis.” The authors may wish to discuss their work in light of the recent publication Ubellacker et al., Nature 2020, which showed the increased sensitivity in cancer cells metastasizing through the blood circulation relative to ones spreading via the lymph node.

Minor suggestions:

1. Per the journal guidelines, the authors may wish to present all data points for the viability curves and bar graphs. Currently many of the error bars (e.g. Figure 1C) are missing and it poses a challenge for properly reading these results.

2. Figure 3E, the differences in fatty acid uptake appear rather small. Are these differences statistically significant?

3. In the title, the authors emphasized “Dysregulated cholesterol homeostasis results in ...”, given the potential ambiguity in the mechanisms of 27-HC action, the authors may consider directly mentioning “Acquired 27-hydroxylcholesterol resistance results in ...” to avoid over-generalization.

4. Line 271-272, “From these data, we concluded that increased lipid uptake, and to a lesser extent lipid biosynthesis, is a feature of cellular resistance to the anti-proliferative actions of 27HC.” At this point, the authors have not demonstrated that increased lipid uptake and biosynthesis are functional contributors of the 27-HC resistance phenotype yet, it may be more accurate to state as “... is a cellular feature of 27HC-resistant cells”.

5. In Ref 82, the contemporary GPX4 inhibitor JKE-1674 introduced in the Eaton et al. Nature Chemical Biology 2020 study show improved stability in mice relative to other GPX4 inhibitors, but the target engagement and therapeutic index in vivo remain to be fully demonstrated. The authors may want to clarify this in their statement.

6. Figure 1E, 3A, 4C, S2A, S4A, S4F, S7B, the authors may wish to provide color keys for the heat maps. The color coding in the current figures are misleading, as in a few figures the fold changes are fairly small, but the color differences are similar as the panels that have much bigger differences.

7. Line 248, a typo: "BIODIPY" should be "BODIPY"

Reviewer #3 (Remarks to the Author); expert on breast cancer and metastasis:

In this manuscript, the authors demonstrate that chronic exposure of ER-negative cancer cells to 27HC results in remodeling of their lipid metabolism and increases the tumorigenic and metastatic capacity of these cells. The authors show that 27HC initially inhibits cell proliferation through negative feedback regulation of the cholesterol/lipid biosynthesis pathways. 27HC resistant cells (27HCR) show less sensitive to this inhibition by upregulating the import of exogenous lipids and cholesterol, and eventually improve the tumorigenesis in vitro and in vivo. Additionally, they show that the formation of cytotoxic lipid peroxides in 27HC resistant cells is accommodated by upregulating GPX4 expression and downregulating the expression of enzymes required for lipid oxidation. They further find that 27HC resistant cells are exceptionally resistant to drugs that directly interfere with GPX4 function or activity. Finally, they demonstrate that GPX4 knockdown attenuates the enhanced metastatic activity of 27HC resistant cells.

The authors present multiple lines of evidence to support their notion. The experiments are well-designed and controls are presented. It should be greatly appreciated that many different ER negative cell models with both human and mouse origination were used in designed studies to ensure the rigor and reproducibility of their research. It is a novel observation to identify chronic exposure of 27HC induces TNBC cells to obtain the enhanced capability in tumor progression. The discovery of linking the ferroptosis with tumor progression is also novel. The major deficiencies of this study are related to the quality of the results presented. Too many inconsistency or even contradicted results obtained from the tested cell models dampened the readers' enthusiasm on the research. Moreover, the rationale for GPX4 as the critical molecule for 27HCR cells is not presented and the preliminary data is not solid to support the author's hypothesis that GPX4 is the major critical factor contributing to the function of 27HCR cells. The detailed information of the related questions is listed as follows.

Major questions:

- 1) It is unclear what condition was used in Figure 1C. CFS or FBS?
- 2) In Figure 2, it is apparent that 27HCR cells promote tumor growth more vigorously than 27HCS cells and the induced metastasis should be tumor growth dependent. In this case, I would suggest the title of the manuscript be changed to "Dysregulated cholesterol homeostasis results in resistance to ferroptosis and increased tumor progression".
- 3) In figure 3D, a statistical analysis should be provided to show 27HCR cells are more sensitive to respond to lipid depletion than 27HCS cells.
- 4) In Figure 3E, it looks like that there is no difference in 4T1 and Py230 cells regarding Fatty acid uptake. No statistical analysis was applied to this study.
- 5) The expression level changes are very limited in most genes examined in Figure 3F. There are very fewer common genes between the three tested cell lines especially between the human cell line and mouse cell lines. Utilization of more cell line models or a genome-wide approach may help to clarify this issue.
- 6) The conclusion that increased expression of GPX4 and/or decreased stress on this pathway through downregulation of genes associated with lipid peroxidation is a characteristic of 27HCR cells

is not convincing based on provided Figure 4C and Supplementary Figure 7. There are a lot of inconsistencies between selected cell lines. For example, for the five examined cell models, only B16F10 cells showed upregulated GPX4 in 27HCR cells compared to 27HCS cells. Therefore, it is hard to believe that GPX4 is the major critical molecule responsible for 27HC resistant cells

7) The conclusion of targeting GPX4 did not change cell viability is not convincing. B16F10 cells showed decrease in cell viability in both 27HCS and 27HCR status. However, no statistical analysis was applied.

8) It is unclear why, to study the effect of GPX4, the author used the mouse xenograft models generated through intravenous injection, but not orthotopic injection. The orthotopic model will provide a chance to clarify whether the loss of GPX4 impacts metastasis specifically or tumor progression in general.

9) The corresponding author was also involved in another publication showing that 27 HC facilitates breast cancer metastasis through its actions on immune cells. A discussion should be provided about whether immune cells also play a role in ER negative cells, what the relationship between these two different mechanisms, and which one may be the major mechanism in current setting.

Minor questions:

1) It is a big concern that statistical analysis is missing for many experiments presented in multiple figures and supplementary figures.

2) "uM" should be "μM".

Response to reviewers

All of the comments from the individual reviewers have been copied verbatim (and bolded) but broken up into sections in some places to separate the issues raised.

Reviewer #1 (Remarks to the Author); expert on lipid metabolism and cancer:

In this manuscript, the authors show that acute treatment with 27-hydroxycholesterol, 27-HC, an intermediate of cholesterol degradation that is elevated in patients with hypercholesterolemia, causes growth inhibition in ER negative breast cancer and metastatic melanoma cells. This is in contrast to their previous findings in ER-positive breast cancer, where 27-HC acts growth promoting by functioning as a partial ligand of the oestrogen receptor. The authors then go on to generate TNBC cells that are made resistant to the effects of 27-HC. While the acute inhibitory effect of the compound on expression of target genes of SREBP and LXR is still intact, the 27-HC resistant populations show an enhanced capacity for tumour growth and metastasis formation. The authors also observe that resistant cell accumulate higher levels of several lipid classes, most likely due to enhanced lipid uptake. As lipid accumulation has previously been linked to ferroptosis, an iron-dependent form of cell death mediated by lipid peroxidation, the authors next investigated whether 27-HC resistant cells show different activity of the ferroptosis machinery. They found that 27-HC resistant cells show differential expression of several genes linked to lipid peroxidation, namely GPX4, ACSL4, LPCAT3 and PEBP1. They also observed increased resistance towards GPX4 inhibitors (RSL3 and ML210) as well as reduced lipid peroxidation in 27-HC resistant cells. Finally, the authors demonstrate that silencing of GPX4 reduces metastasis formation by B16F10 melanoma and Py230 breast cancer cells. The authors conclude that chronic exposure to 27-HC selects for cells that show an enhanced resistance towards ferroptotic cell death and thus promotes the ability of cancer cells for metastatic outgrowth.

This study contains a large amount of data generated across several cellular systems, as 5 breast and two melanoma cell lines were used for most of the experiments. It describes a very interesting concept of selection during chronic exposure to a compound that is found in the circulation of patients with hypercholesterolemia, thus elucidating the mechanism behind the observed increased cancer risk. While most experiments are very clear and adequately performed and the conclusions are sound, there is some lack of clarity in the underlying concept that should be addressed before publication.

- We thank the reviewer for his/her positive opinion of the study and for acknowledging the large body of work that the study describes.

Specific issues that the reviewer requested that we address:

1. The main point of criticism is that the authors show that 27-HC resistant cells accumulate lipids, most likely due to enhanced uptake, which makes them vulnerable to lipid peroxidation. They then hypothesize that pathways that protect against lipid peroxidation, i.e. GPX4, should be upregulated to support survival in resistant cells. Alternatively, enzymes that promote lipid peroxidation can be downregulated. They therefore investigate the expression of GPX4 and three enzymes that have previously been shown to be required for lipid peroxidation (i.e. ACSL4 and AGPAT3, which are required for the incorporation of poly-unsaturated fatty acids into membrane lipids, and PEBP1 that mediates the access of lipoxygenases to membrane lipids) in the

panel of 27-HC sensitive and resistant cells. However, the differential expression of these factors varies widely across the cell lines used. For example, 27HC resistant HCC1954 cells show downregulation of GPX4, while ACSL4 is upregulated. Nevertheless, the cells show a marked resistance towards the GPX4 inhibitors and erastin. In contrast, B16F10 melanoma cells upregulate all of the tested factors but still show decreased sensitivity towards ferroptosis induction.

- Although the reviewer did not request a response to this comment, it highlighted a lack of clarity in our message/conclusion. In short, we propose that cells, in responding to the stress (increased production of lipid peroxides) imposed by 27HC-induced lipid uptake, expose a vulnerability that they respond to by either (a) increasing the expression/activity of GPX4 and/or (b) by downregulating the expression of mRNAs encoding enzymes involved in lipid peroxidation. We have in our rewriting of the manuscript brought more clarity to this issue.
2. **Given this lack of correlation, it is possible that the altered expression of the investigated factors is not, or not solely, responsible for ferroptosis resistance. Instead, the observed difference in ferroptosis sensitivity could actually be caused by alterations in lipid composition, which has previously been linked to ferroptosis sensitivity. While the authors have compared overall lipid levels between sensitive and resistant cells, they did not specify changes in PUFA-containing lipids.**
- We performed a comprehensive lipidomics analysis comparing HCC1954 and its 27HCR derivative. Whereas the total lipid content was higher in the HCC1954-27HCR cells, we did not find a particular enrichment in the saturated, monounsaturated or polyunsaturated fatty acid containing phospholipids. However, we noted that the levels of total monounsaturated and oleic acid containing neutral lipids are significantly higher in 27HCR cells when compared to 27HCS cells. These results are now highlighted in Supplementary Table 3. In addition, we found that several of the fatty acid transporters that are responsible for cellular uptake of oleic acid, (i.e. ACSL3, SLC27A1, SLC27A2, SLC27A3 and SLC27A5), are significantly up-regulated in HCC1954-27HCR cells compared to -27HCS cells (Supplementary Table 4). As pointed out by the reviewer, Ubellacker et al. demonstrated that triacylglycerols with a high oleic acid content protect cancer cells from ferroptosis; a finding that is consistent with our data. We have highlighted (and referenced) this published paper and included a discussion of the re-analysis of our lipidomics data in the revised manuscript.
3. **Another possible mechanism could involve the activation of NRF2 and the corresponding induction of processes that protect against oxidative stress, for example through increased production of GSH and NADPH. This possibility could also be investigated.**

We profiled the expression of NRF2 regulated genes that have been shown by others to be involved in ferroptosis (GPX4, SLC7a11, GCLC, GCLM, GSS, NQO1, MT1 (and NRF2)). This data is included in Supplementary Table 5. There was no consistent pattern in the expression of these NRF-2 related genes across cell lines. However, in the revised manuscript we discuss the possible involvement of NRF2 in the resistance phenotype and acknowledge that further work should be performed to define the importance of this pathway.

4. **Nevertheless, the authors show very convincingly that depletion of GPX4 reduces the metastatic capacity of the two cancer cell lines studied. In the absence of a clearer**

mechanism mediating 27-HC resistance, the results should be more cautiously discussed.

- We hope that the reviewer agrees that we have, in the revised manuscript, suggested alternative explanations/interpretations of our data and that we have highlighted the issues that remain to be resolved.

Below are also some more specific issues that should be addressed.

1. It should be specified whether the 27-HC treatment shown in Fig 1C and S1A was done in delipidated medium.

- Experiments for Fig.1c and Supplementary Fig.1a were conducted in normal serum and lipid-containing media. We have modified the text to make this clearer in the revised manuscript.

2. The conclusion that 27-HC resistant cells activate lipid synthesis is not supported by the data. Accumulation of TAGs could also be the consequence of reduced lipid oxidation.

- As suggested by the reviewer, we tested whether alterations in lipid oxidation in 27HCR cells could explain the increased lipid accumulation (and in this manner contribute to 27HC resistance). However, we found that treating cells with etomoxir (an inhibitor of fatty acid β -oxidation) does not alter total lipid content in these cells (Supplementary Fig5d). We have mentioned the fact that changes in fatty acid oxidation may be important in some cells/contexts.

3. The increased signal of Bodipy dye to demonstrate enhanced lipid uptake shown in Figure 3E is very small for 4T1 and Py230 cells. Is this difference significant?

- We agree with the reviewer that lipid uptake, as determined by BODIPY FL-C16, does not appear to be strongly increased in the 4T1 and Py230 models as it is in the other models studied. It is possible that the bulk lipid uptake is a surrogate for the uptake of subspecies of lipid that imparts the resistance phenotype in these cells (note that in these cells, as in others studied, the 27HCR phenotype is only manifest in lipid containing media). We have added this discussion in the revised manuscript.

4. The investigation of different inhibitors towards components of the lipid uptake/remodelling machinery shown in Figure S6 is not very clear.

- In re-reviewing the data we presented on this topic in Figure S6, we agree that it is exceptionally complicated. Further, since the study did not allow us to identify a single mechanism to explain the increased lipid uptake, it adds little to the paper in its current form. Thus, we have removed the figure in the supplementary data section of the paper. The text in the paper did not need modification.

5. Line 329. Erastin is an inhibitor of xCT and blocks GSH synthesis rather than downregulates GPX4. This should be clarified in the text.

- We agree that erastin is an inhibitor of xCT, however, it has also been shown to promote GPX4 degradation (Wu, Z. *et al.* Chaperone-mediated autophagy is involved in the execution of ferroptosis. *Proc Natl Acad Sci U S A* **116**, 2996-3005 (2019)). It is more appropriate to say that treatment with erastin results in a downregulation of GPX4 expression. We have reworded the text accordingly.
- 6. Line 341: More rationale should be provided why only ACSL4, LPCAT3 and Pebp1 were investigated as potential mediators of lipid peroxidation.**
- We actually profiled the expression of a large panel of genes implicated in ferroptosis and found that depending on the cell lines, different aspects of the pathway were altered (now included as Supplementary Table 5). However, we focused our studies on ACSL4, LPCAT3 and PEBP1 as (a) they have been implicated in the formation of lipid peroxides and (b) they were the most commonly regulated among the models we were studying. We also found increased expression of the xCT transporter gene, SLC7A11, and downregulation of genes encoding proteins involved in iron metabolism. Their potential involvement in the observed phenotypes are discussed in the text.
- 7. Line 370: A link between ferroptosis resistance and metastasis formation has been described before (Alvarez et al Nature 2017 and Ubellacker et al. Nature 2020). These references could be included in a revised version of the manuscript.**
- Thank you for bringing these manuscripts to our attention. We have included these references in the revised manuscript.
- 8. The display of the qPCR data as heatmaps (Fig. 1E, 3A and 4C) is quite unusual and makes the data hard to interpret. Do the panels represent biological replicates from the same pool of sensitive or resistant cells or independently generated pools/clones of resistant populations? This should be clarified.**
- We recognize the issue raised by the reviewer and to assist interpretation we have now included detailed qPCR results (i.e. fold change, SD, p-value) in our Supplementary Tables. The data presented are technical replicates (the same pool of sensitive or resistant cells plated in individual plates) of the analysis of mRNA from a single study. We repeated key phenotypes in independent biological replicate experiments (now indicated in Resource Data file)

Reviewer #2 (Remarks to the Author); expert on ferroptosis and cancer:

In the present manuscript, Liu et al. reports that in contrast to the pro-tumorigenic role of 27-hydroxycholesterol (27-HC) in ER-positive breast cancer cells, 27-HC treatment attenuates the proliferative and migratory potential of ER-negative breast cancer cells. Additionally, these cancer cell line models can acquire resistance to 27-HC under chronic selection. Importantly, the 27-HC-resistant (27HCR) cells appear to exhibit higher tumorigenic and metastasis formation potential than their non-selected, parental ER-negative breast cancer cells. 27HCR cells also display higher levels of lipid accumulation, and decreased sensitivity to GPX4 inhibition induced ferroptosis – both features may be relevant contributors to the increased tumorigenesis and metastasis capacity of the 27HCR cells.

Overall, the results in this work are both interesting and can be important. The findings related to 27-HC resistance are novel and of interest to the cancer research community. The data are largely consistent across multiple cell line models that were assessed in this study, therefore are largely convincing. Together, the work is worth considering for publication at Nature Communications, given that the following concerns are properly addressed:

- We thank the reviewer for the positive opinion of our work.

Major questions:

1. The authors have proposed that the upregulation of lipid receptors are crucial to increased lipid accumulation in 27HCR cells, and showed that knockdown the expression of a single lipid transporter vLDLR is not sufficient to suppress lipid accumulation. Does overexpression of vLDLR, FABP4 or CD36 at least partially recapitulate the lipid accumulation phenotype in the specific ER-negative cancer cells used, and how does the expression of these genes alter the sensitivity to ferroptosis?

- As suggested by the reviewer we overexpressed CD36 in (A) HCC1954 and (B) MDAMB436 breast cancer cells (Figure a). Interestingly, we did not observe an increase in total lipid content or lipid uptake in the CD36 over-expressing cells (Figure b). However, we did show that overexpression of CD36 protects cells against ferroptosis inducing agents (Figure c). It is possible that CD36 facilitates the uptake of a specific lipid that is causally responsible for

the resistance phenotype. We have included this data for the reviewer only as, although it is supportive of our general thesis, it is clear that a considerable amount of additional work needs to be done to dissect the implicated pathways.

- 2. In this referee's opinion, the logic of the experiments described in the section "27HC resistant cells are protected from ferroptotic cell death" and Figure 4 can be optimized: When there is increased lipid accumulation, a plausible assumption is that cancer cells may experience increased lipid oxidative stress and consequently alterations in the sensitivity to GPX4 inhibition-induced ferroptosis. Lipid accumulation is often insufficient to trigger the speculation that cells would compensatorily downregulate lipid repair enzymes like GPX4 or lipid biosynthesis genes. (Also a minor point, ACSL4 and LPCAT3 are not genes required for lipid peroxidation, they are enzymes required for the biosynthesis of polyunsaturated phospholipids, which are substrates of lipid peroxidation.) (Alternatively, the authors may consider introducing their characterizations of the GPX4 inhibitor sensitivity in the 27HCS and 27HCR cells before their investigation into GPX4 expression levels.)**
 - The flow of the story that the reviewer suggested is what we were trying to achieve but clearly we did not execute. We hope that the revised manuscript makes it clear that we conclude that sensitivity to GPX4 inhibitors can be accomplished by either increased expression of GPX4 itself (Figure 4a, b) and/or through downregulation of enzymes responsible for the production of PUFAs (and lipid peroxides). We posit that regardless of the upstream events GPX4 is a convergent node of such regulation. We hope that the streamlined text flows/reads better.
- 3. Related to this issue, in line 336-337, the authors stated that "These data suggest that the synthesis of GPX4 is dysregulated in 27HCR cells in favor of increased expression of GPX4." This conclusion appears largely unsupported by the data in Figure 4C and Supplementary Figure 7, since the GPX4 expression level in 27HCR does not seem to be higher than that in the 27HCS cells (at least in the absence of 27HC treatment, which is the same condition as the tumor implantation and metastasis assays). The authors may want to clarify this issue.**
 - We believe that the modifications we have made to address the point above will also address the apparent discrepancy in the data highlighted by the reviewer.
- 4. In this referee's opinion and according to the lipidomics result, the increased resistance to ferroptosis in the GPX4 inhibitors in 27HCR cells are likely at least partially caused by the increased accumulation of lipids containing saturated or monounsaturated fatty acids, and the relatively limited accumulation of polyunsaturated phospholipids (true substrates for lipid peroxidation and ferroptosis). Note that increased levels of monounsaturated fatty acids are inhibitory of ferroptosis sensitivity (Magtanong et al., Cell Chemical Biology 2019), which is further demonstrated in a more recent study (Ubellacker et al., Nature 2020).**
 - As the reviewer points out the lipidomics analysis we performed, comparing HCC1954 and its 27HCR derivative, while showing that the total lipid content was higher in the HCC1954-27HCR did not find a particular enrichment in the saturated, monounsaturated or polyunsaturated fatty acid containing phospholipids. However, we noted that the levels of total monounsaturated and oleic acid containing neutral lipids are significantly higher in

27HCR cells when compared to 27HCS cells. These results are shown in Supplementary Table 3. In addition, we found that several of the fatty acid transporters that are responsible for cellular uptake of oleic acid (i.e. ACSL3, SLC27A1, SLC27A2, SLC27A3 and SLC27A5) are significantly up-regulated in HCC1954-27HCR cells compared to -27HCS cells (Supplementary Table 4). As pointed out by the reviewer, Ubellacker et al. demonstrated that triacylglycerols with a high oleic acid content protects cancer cells from ferroptosis. We have highlighted (and referenced) this published paper and the re-analysis of our lipidomics data in the revised manuscript.

5. Line 406-407, “These data establish an essential role for GPX4 in preventing ferroptosis, an activity that has heretofore not been linked to metastasis.” The authors may wish to discuss their work in light of the recent publication Ubellacker et al., Nature 2020, which showed the increased sensitivity in cancer cells metastasizing through the blood circulation relative to ones spreading via the lymph node.

- We have discussed this excellent publication in the revised manuscript as suggested and have removed the statement that refers to the primacy of our observation.

Minor suggestions:

1. Per the journal guidelines, the authors may wish to present all data points for the viability curves and bar graphs. Currently many of the error bars (e.g. Figure 1C) are missing and it poses a challenge for properly reading these results.

- The data presented is from one experiment with 3-5 technical replicates as representative data from multiple independent repeats, error bars are actually not missing but very tight (now indicated in Resource data file).

2. Figure 3E, the differences in fatty acid uptake appear rather small. Are these differences statistically significant?

- We agree with the reviewer that lipid uptake, as determined by BODIPY FL-C16, does not appear to be strongly increased in the 4T1 and Py230 models. However, other (specific) lipids and/or different aspects of lipid metabolism could be responsible for the increased lipid accumulation in the 4T1 and Py230 models. We have added this discussion in the revised manuscript.

Given that the expression of multiple lipid uptake genes are increased in the 27HCR cells and that uptake of exogenous lipids is required for the pathobiology manifest by 27HCR cells, provides a compelling case that lipid accumulation/dysregulated lipid homeostasis contributes the phenotypes we have observed.

3. In the title, the authors emphasized “Dysregulated cholesterol homeostasis results in ...”, given the potential ambiguity in the mechanisms of 27-HC action, the authors may consider directly mentioning “Acquired 27-hydroxylcholesterol resistance results in ...” to avoid over-generalization.

- In making the statement “...*potential ambiguity in the mechanisms of 27-HC action*”, we believe that the reviewer is referring to the fact that we have shown in other publications that 27HC can function as a modulator of estrogen receptor (ER) function and also modulates

the function of gamma-delta-T-cells in the metastatic niche. However, in this study we are specifically using ER-negative breast cancer cells, did not further treat mice with 27HC in tumor growth and metastasis studies, and have confirmed our findings in both immune competent and immunocompromised mice. Therefore we are confident that the phenotypes we are studying can be attributed to dysregulated cholesterol homeostasis (ie an adaptive response to 27HC mediated inhibition of cholesterol synthesis and uptake). We would ask the reviewers indulgence and allow us to use the modified title we have selected.

4. **Line 271-272, "From these data, we concluded that increased lipid uptake, and to a lesser extent lipid biosynthesis, is a feature of cellular resistance to the anti-proliferative actions of 27HC." At this point, the authors have not demonstrated that increased lipid uptake and biosynthesis are functional contributors of the 27-HC resistance phenotype yet, it may be more accurate to state as "... is a cellular feature of 27HC-resistant cells".**
 - We have changed our description of the results as suggested.
5. **In Ref 82, the contemporary GPX4 inhibitor JKE-1674 introduced in the Eaton et al. Nature Chemical Biology 2020 study show improved stability in mice relative to other GPX4 inhibitors, but the target engagement and therapeutic index in vivo remain to be fully demonstrated. The authors may want to clarify this in their statement.**
 - We thank to reviewer's comments and have clarified this in the main context.
6. **Figure 1E, 3A, 4C, S2A, S4A, S4F, S7B, the authors may wish to provide color keys for the heat maps. The color coding in the current figures are misleading, as in a few figures the fold changes are fairly small, but the color differences are similar as the panels that have much bigger differences.**
 - We have provided the color keys for the heat maps and the data is also included in new Supplementary Tables.
7. **Line 248, a typo: "BIODIPY" should be "BODIPY"**
 - We have corrected it in the revised manuscript.

Reviewer #3 (Remarks to the Author); expert on breast cancer and metastasis:

In this manuscript, the authors demonstrate that chronic exposure of ER-negative cancer cells to 27HC results in remodeling of their lipid metabolism and increases the tumorigenic and metastatic capacity of these cells. The authors show that 27HC initially inhibits cell proliferation through negative feedback regulation of the cholesterol/lipid biosynthesis pathways. 27HC resistant cells (27HCR) show less sensitive to this inhibition by upregulating the import of exogenous lipids and cholesterol, and eventually improve the tumorigenesis in vitro and in vivo. Additionally, they show that the formation of cytotoxic lipid peroxides in 27HC resistant cells is accommodated by upregulating GPX4 expression and downregulating the expression of enzymes required for lipid oxidation. They further find that 27HC resistant cells are exceptionally resistant to drugs that directly interfere with GPX4 function or activity. Finally, they demonstrate that GPX4

knockdown attenuates the enhanced metastatic activity of 27HC resistant cells.

The authors present multiple lines of evidence to support their notion. The experiments are well-designed and controls are presented. It should be greatly appreciated that many different ER negative cell models with both human and mouse origination were used in designed studies to ensure the rigor and reproducibility of their research. It is a novel observation to identify chronic exposure of 27HC induces TNBC cells to obtain the enhanced capability in tumor progression. The discovery of linking the ferroptosis with tumor progression is also novel. The major deficiencies of this study are related to the quality of the results presented. Too many inconsistencies or even contradicted results obtained from the tested cell models dampened the readers' enthusiasm on the research. Moreover, the rationale for GPX4 as the critical molecule for 27HCR cells is not presented and the preliminary data is not solid to support the author's hypothesis that GPX4 is the major critical factor contributing to the function of 27HCR cells. The detailed information of the related questions is listed as follows.

- We thank the reviewer for his/her positive comments and for recognizing the large amount of work that went into this study.

Major questions:

1. It is unclear what condition was used in Figure 1C. CFS or FBS?

- We used FBS for all of the studies in Figure 1c and this has been clarified.

2. In Figure 2, it is apparent that 27HCR cells promote tumor growth more vigorously than 27HCS cells and the induced metastasis should be tumor growth dependent. In this case, I would suggest the title of the manuscript be changed to "Dysregulated cholesterol homeostasis results in resistance to ferroptosis and increased tumor progression".

- We have changed the title as suggested and it now reads *"Dysregulated cholesterol homeostasis results in resistance to ferroptosis increasing tumorigenicity and metastasis in cancer"*

3. In figure 3D, a statistical analysis should be provided to show 27HCR cells are more sensitive to respond to lipid depletion than 27HCS cells.

- We have included a statistical analysis in the revised figures as requested.

4. In Figure 3E, it looks like that there is no difference in 4T1 and Py230 cells regarding Fatty acid uptake. No statistical analysis was applied to this study.

- We agree with the reviewer that lipid uptake, as determined by BODIPY FL-C16, does not appear to be strongly increased in the 4T1 and Py230 models. However, other (specific) lipids and/or different aspects of lipid metabolism could be responsible for the increased lipid accumulation in the 4T1 and Py230 models. We have added this discussion in the revised manuscript.

Given that the expression of multiple lipid uptake genes are increased in the 27HCR cells and that uptake of exogenous lipids is required for the pathobiology of the 27HCR cells, this provides a compelling case that lipid accumulation/dysregulated lipid homeostasis contributes the phenotypes we have observed.

- 5. The expression level changes are very limited in most genes examined in Figure 3F. There are very fewer common genes between the three tested cell lines especially between the human cell line and mouse cell lines. Utilization of more cell line models or a genome-wide approach may help to clarify this issue.**

We interpret the differences in the degree of expression of the various “ferroptosis-linked genes” in different cell models to reflect the fact that resistance to 27-HC induced lipid uptake/peroxidation can occur through the disruption of many parts of the pathway. Ultimately, however, the effect is to reduce the pressure on the GPX4 pathway. Dissecting the relative importance of individual enzymes in the resistance phenotype is a subject of a new project that we are embarking on.

- 6. The conclusion that increased expression of GPX4 and/or decreased stress on this pathway through downregulation of genes associated with lipid peroxidation is a characteristic of 27HCR cells is not convincing based on provided Figure 4C and Supplementary Figure 7. There are a lot of inconsistencies between selected cell lines. For example, for the five examined cell models, only B16F10 cells showed upregulated GPX4 in 27HCR cells compared to 27HCS cells. Therefore, it is hard to believe that GPX4 is the major critical molecule responsible for 27HC resistant cells.**

In re-reading our manuscript it was clear that we did not do a good job of summarizing and presenting our interpretation of the data in Figure 4. We were definitely influenced by the fact that we knew the data in Figure 5 which specifically looked at the role of GPX4 in the biology of the 27HCR cells. As shown, all of the 27HCR cells are substantially more resistant to direct inhibitors of GPX4 (RSL3 and ML210) and to erastin (which decreases intracellular GSH). As the reviewer points out the 27HCR phenotype can only be attributed to increased GPX4 expression in a few of the models we developed. In others we see a dramatic downregulation of the expression of the enzymes (ACSL4, LPCAT3 and PEBP1) that are responsible for the production of PUFAs (substrates for lipid peroxidation) and/or increased expression of SLC7A11 (a xCT transporter gene). Together these activities reduce the cellular requirement for GPX4 and this manifests as resistance to GPX4 inhibitors. It was the likely convergence of all of the activities on GPX4 that led us to the definitive studies we present in Figure 5.

- 7. The conclusion of targeting GPX4 did not change cell viability is not convincing. B16F10 cells showed decrease in cell viability in both 27HCS and 27HCR status. However, no statistical analysis was applied.**

- We determined whether GPX4 KD regulates cell proliferation using a DNA content assay. As shown in Supplementary Fig. 7b, GPX4 KD in B16F10-27HCS cells delayed cell growth, and moderately inhibited the growth of B16F10-27HCR cells. These results are consistent with a recent publication (Ubellacker et al., *Nature* 2020) showing that deletion of GPX4 did not significantly affect the proliferation or survival of mouse melanoma cells *in vitro*. We have added statistical analysis to the figures and revised the conclusion.

8. **It is unclear why, to study the effect of GPX4, the author used the mouse xenograft models generated through intravenous injection, but not orthotopic injection. The orthotopic model will provide a chance to clarify whether the loss of GPX4 impacts metastasis specifically or tumor progression in general.**
 - We performed the requested experiment. In revised manuscript, we provided data showing that knockdown of GPX4 inhibited the primary tumor growth only in 27HCR- but not 27HCS-B16F10 model. This result indicates that the metabolic reprogramming of tumor cells by chronic exposure to 27HC exposes a reliance on GPX4 in both primary tumor and metastatic growth.
9. **The corresponding author was also involved in another publication showing that 27 HC facilitates breast cancer metastasis through its actions on immune cells. A discussion should be provided about whether immune cells also play a role in ER negative cells, what the relationship between these two different mechanisms, and which one may be the major mechanism in current setting.**

We believe both tumor-intrinsic and -extrinsic mechanisms are responsible for the increased malignancy (tumor growth and metastasis) in 27HCR cells. The current study focused on determining how tumor cells adapt to chronic long-term 27HC exposure whereas the previous work studies explored cellular and systematic response to acute 27HC treatment. Note that although the 27HC resistant cells were derived through chronic exposure to 27HC, the oxysterol was not required to maintain the phenotype and none of the *in vivo* work presented required the administration of 27HC. In the study the reviewer refers to (and which we discuss in the current manuscript) the phenotype observed results from a “preconditioning” of the metastatic niche by administering 27HC to the mice before introduction of the cancer cells). The mechanisms are not mutually exclusive but please note that the increased metastasis of 27HCR cells we have observed in the current study is manifest in both immune competent and immune compromised mice.

Minor questions:

1. **It is a big concern that statistical analysis is missing for many experiments presented in multiple figures and supplementary figures.**
 - We have included additional statistical analyses in revised manuscript as requested.
2. **“uM” should be “μM”.**
 - We have corrected this mistake in the revised manuscript.

REVIEWER COMMENTS

Reviewer #1 (Remarks to the Author):

Overall, it has to be commented that the authors did not consider some of the valid criticism, and there are a couple of important points that have not been addressed despite being raised by more than one reviewer. The revised manuscript, while still showing important and interesting data, remains difficult to read and the line of argument is hard to follow.

For example, the authors did not change the display of their qPCR data. As colour scales have now been added, it is revealed that the heatmaps are coloured by row min/max. This over-emphasizes potentially small differences, making comparisons between genes and cell lines almost impossible. In addition, some statistical analyses are only shown in the supplementary tables, making it even harder to evaluate the solidity of the findings. If the authors want to keep the heatmap display, they should be at least coloured uniformly, e.g. by log₂ fold change.

There is also still an issue with the lipidomics analysis represented in supplementary figure 4f and supplementary table 3. The authors state in the text that MUFA-containing TAGs and DAGs were among the most significantly upregulated lipid species (see line 267). However, the heatmap also shows substantial upregulation of fully saturated TAGs, in accordance with an overall induction of neutral lipid abundance, as suggested by the increased formation of lipid droplets. Moreover, the graphs displayed next to this table show no obvious difference in the saturation in total lipids and PEs. The authors should provide some level of analysis/display of their data that would not require the reader to look up individual lipid species in the supplementary tables to support their conclusions. In addition, data supporting the statement in line 270 that “levels of total monounsaturated fatty acid and oleic acid containing neutral lipids are significantly higher in 27HCR cells” is missing. One way to display lipidomics data could be to show a volcano plot (plotting log₂ fold change versus FDR-corrected p-value) with MUFA-containing DAGs and TAGs indicated in different colours. Specific lipid species that support essential conclusions should be displayed separately.

Another issue that still needs to be improved is the discussion of the data shown in Figure 4c, particularly as the inconsistencies presented by these data were noticed by two reviewers. While the authors state in their rebuttal that the discussion of the data in this figure was informed by the functional analysis of GPX4 inhibition presented later, it nevertheless represents an inconsistency that should be resolved. The small number of genes investigated obviously does not provide a comprehensive assessment of all mechanisms that contribute to lipid peroxidation or oxidative stress production/removal in sensitive and resistant cells. In the light of this limitation and the lack of a clear correlation of the four genes that were investigated, the data should either be removed altogether or discussed much more carefully in the text. The same also applies to the evaluation of GPX4 protein levels in sensitive and resistant cells (shown in Fig. S6a). Here, some of the resistant cell lines show a clear downregulation of GPX4 expression when compared to their sensitive counterparts, which does not fit to the model of enhanced stress resistance implied by the text.

Reviewer #2 (Remarks to the Author):

The authors have addressed this referee's concerns satisfactorily.

Minor:

Line 895, there is a typo in SREBP2.

Reviewer #3 (Remarks to the Author):

In this revised manuscript, the authors investigate how chronic exposure of ER-negative cancer cells to 27HC results in remodeling their lipid metabolism and increases their tumorigenic and metastatic capacity. The discovery is novel and experimental evidence provided is solid after revision. I am recognizing that the authors have been working hard and put tremendous effort into this revision. I agree that most of my questions or concerns were addressed and discussed properly.

Response to reviewers

All of the comments from the individual reviewers have been copied verbatim (and bolded) but broken up into sections in some places to separate the issues raised.

Reviewer #1 (Remarks to the Author); expert on lipid metabolism and cancer:

Overall, it has to be commented that the authors did not consider some of the valid criticism, and there are a couple of important points that have not been addressed despite being raised by more than one reviewer. The revised manuscript, while still showing important and interesting data, remains difficult to read and the line of argument is hard to follow.

For example, the authors did not change the display of their qPCR data. As colour scales have now been added, it is revealed that the heatmaps are coloured by row min/max. This over-emphasizes potentially small differences, making comparisons between genes and cell lines almost impossible. In addition, some statistical analyses are only shown in the supplementary tables, making it even harder to evaluate the solidity of the findings. If the authors want to keep the heatmap display, they should be at least coloured uniformly, e.g. by log₂ fold change.

- We thank the reviewer for the suggestion and in response we have replaced all of the heatmaps with bar graphs as a mean to present the relevant qPCR data. Having done this we agree that the data is much easier to visualize and understand.

There is also still an issue with the lipidomics analysis represented in supplementary figure 4f and supplementary table 3. The authors state in the text that MUFA-containing TAGs and DAGs were among the most significantly upregulated lipid species (see line 267). However, the heatmap also shows substantial upregulation of fully saturated TAGs, in accordance with an overall induction of neutral lipid abundance, as suggested by the increased formation of lipid droplets. Moreover, the graphs displayed next to this table show no obvious difference in the saturation in total lipids and PEs. The authors should provide some level of analysis/display of their data that would not require the reader to look up individual lipid species in the supplementary tables to support their conclusions. In addition, data supporting the statement in line 270 that “levels of total monounsaturated fatty acid and oleic acid containing neutral lipids are significantly higher in 27HCR cells” is missing. One way to display lipidomics data could be to show a volcano plot (plotting log₂ fold change versus FDR-corrected p-value) with MUFA-containing DAGs and TAGs indicated in different colours. Specific lipid species that support essential conclusions should be displayed separately.

- As suggested by the reviewer we have now included volcano plots as a means to present the changes in all of the lipid species in 27HCR cells compared to 27HCS cells. The lipids of particular interest are labelled in different colors (**Supplementary Fig. 4f and g**). Neutral lipids are the most significantly increased lipid species in 27HCR cells, of which MUFA- containing neutral lipids are the most abundant (70% of total neutral lipid). Saturated FA-containing neutral lipids represent about 20% of the total neutral lipids (**Supplementary Fig. 4h and 4i in revised manuscript**). Presentation of the data in this manner highlighted the fact that the total MUFA-containing and oleic acid containing- neutral lipids, but not saturated neutral lipids,

were increased in 27HCR compared to 27HCS cells (**Supplementary Fig. 4j**). The text has been revised to reflect these new insights.

Another issue that still needs to be improved is the discussion of the data shown in Figure 4c, particularly as the inconsistencies presented by these data were noticed by two reviewers. While the authors state in their rebuttal that the discussion of the data in this figure was informed by the functional analysis of GPX4 inhibition presented later, it nevertheless represents an inconsistency that should be resolved. The small number of genes investigated obviously does not provide a comprehensive assessment of all mechanisms that contribute to lipid peroxidation or oxidative stress production/removal in sensitive and resistant cells. In the light of this limitation and the lack of a clear correlation of the four genes that were investigated, the data should either be removed altogether or discussed much more carefully in the text.

We hope we have addressed this issue by presenting the mRNA expression data as bar graphs rather than in the form of a heatmap. When presented in this manner it is clear that the expression of the implicated genes is downregulated in the 27HCR derivatives of Py230, HCC1954 and MDAMB436 H cells. The exception is the B16F10 cells, however, as noted in the text this cell line may rely more on increased GPX4 expression. We again thank the reviewer for steering us away from using heatmaps to present these data!

The same also applies to the evaluation of GPX4 protein levels in sensitive and resistant cells (shown in Fig. S6a). Here, some of the resistant cell lines show a clear downregulation of GPX4 expression when compared to their sensitive counterparts, which does not fit to the model of enhanced stress resistance implied by the text.

We have revised the text to more clearly state that the major difference we observe is that in 27HCS cells we see a downregulation of GPX4 in response to treatment 27HC (or erastin). This downregulation is not observed in the 27HCR cells. We allude to the fact that this may relate to changes in the stability of the GPX4 protein, something we will definitely explore in the future.

Reviewer #2 (Remarks to the Author); expert on ferroptosis and cancer:

The authors have addressed this referee's concerns satisfactorily.

Minor:

Line 895, there is a typo in SREBP2.

- We thank the referee for the positive opinion of the revised manuscript. We have changed the typographical error (as requested) in the revised manuscript.

Reviewer #3 (Remarks to the Author); expert on breast cancer and metastasis:

In this revised manuscript, the authors investigate how chronic exposure of ER-negative cancer cells to 27HC results in remodeling their lipid metabolism and increases their tumorigenic and metastatic capacity. The discovery is novel and experimental evidence provided is solid after revision. I am recognizing that the authors have been working hard

and put tremendous effort into this revision. I agree that most of my questions or concerns were addressed and discussed properly.

- We thank the reviewer for the comments, and we are glad to be able to address all the issues.

REVIEWER COMMENTS

Reviewer #1 (Remarks to the Author):

The authors have addressed all points raised. The manuscript is now suitable for publication.